# FaLW: A Forgetting-aware Loss Reweighting for Long-tailed Unlearning

**Liheng Yu**[1,3]**, Zhe Zhao**[1,3]**, Yuxuan Wang**[1]**, Pengkun Wang**[1,2,*]**Xiaofeng Cao** [4]**,**
**Binwu Wang**[1,2]**, Yang Wang**[1,2,*]
[1]University of Science and Technology of China(USTC)
[2]Suzhou Institute for Advanced Research, USTC
[3]City University of Hong Kong
[4]Tongji University
{yuliheng, zz4543, xuco, wdcxy}@mail.ustc.edu.cn,
xiaofeng.cao.uts@gmail.com, pengkun@ustc.edu.cn

## Abstract

Machine unlearning, which aims to efficiently remove the influence of specific data from trained models, is crucial for upholding data privacy regulations like the "right to be forgotten". However, existing research predominantly evaluates unlearning methods on relatively balanced forget sets. This overlooks a common real-world scenario where data to be forgotten, such as a user's activity records, follows a long-tailed distribution. Our work is the first to investigate this critical research gap. We find that in such long-tailed settings, existing methods suffer from two key issues: *Heterogeneous Unlearning Deviation* and *Skewed Unlearning Deviation*. To address these challenges, we propose FaLW, a plug-and-play, instance-wise dynamic loss reweighting method. FaLW innovatively assesses the unlearning state of each sample by comparing its predictive probability to the distribution of unseen data from the same class. Based on this, it uses a forgetting-aware reweighting scheme, modulated by a balancing factor, to adaptively adjust the unlearning intensity for each sample. Extensive experiments demonstrate that FaLW achieves superior performance. Code is available at **Supplementary Material**.

## 1 Introduction

With the enactment of regulations such as the General Data Protection Regulation (GDPR), the "right to be forgotten" (Regulation, 2018; Hoofnagle et al., 2019) has become a cornerstone of data privacy protection. In artificial intelligence, the right's realization has given rise to emerging technical paradigms of machine unlearning. Its core objective is to accurately and efficiently remove the influence of specific training samples from a pre-trained ML model (Shaik et al., 2024).

Currently, machine unlearning methods can be broadly categorized into two main types: exact unlearning (Guo et al., 2019) and approximate unlearning (Izzo et al., 2021). The former primarily focuses on methods that provide provable guarantees or certified removal. Within this category, retraining from scratch is considered the gold standard (Thudi et al., 2022b). This approach involves training a new model from scratch on the retain set, which is the original dataset minus the data to be forgotten. However, such retraining-based approaches are computationally prohibitive and thus impractical for large-scale models prevalent today.

In contrast, approximate unlearning presents a practical alternative, aiming for fast and effective removal of data influence. While these methods may not always come with formal guarantees, their efficacy is typically evaluated through empirical metrics, such as Membership Inference Attacks (MIA) (Carlini et al., 2022), and they often operate under fewer constraints on the data, model, or algorithm compared to their exact counterparts. To date, the evaluation of machine unlearning has

---

*Corresponding author.

predominantly centered on image classification tasks, where, typically, the forget set is constructed by randomly sampling a subset of the original training data.

However, we empirically observe (Figure 1(a)) that *random sampling consistently yields forget sets that are nearly balanced. This stands in stark contrast to many real-world scenarios, where forget requests are often naturally long-tailed.* For instance, when a user deactivates their online shopping account (Balasubramanian et al., 2024), their data, shaped by personal interests, is typically concentrated in a few specific categories. Based on this, machine unlearning under such long-tailed distributions remains largely unexplored. To bridge this gap, we introduce the concept of *Unlearning Deviation* to quantify the unlearning outcome for individual samples. Through analysis on imbalanced forget sets, we uncover two critical phenomena:

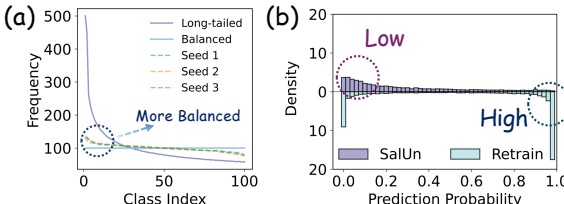

Figure 1: (a) Distributions of a **20% CIFAR-100** forget set generated via **balanced, long-tailed, and random sampling** (Seeds 1-3 denote different runs). (b) Predicted probability distributions for SalUn vs. Retrain (ResNet-18) after unlearning a 20% randomly sampled forget set from **CIFAR-100**.

❶ *Heterogeneous Unlearning Deviation:* models tend to over-forget some samples while under-forgetting others.

❷ *Skewed Unlearning Deviation:* the magnitude of this deviation is disproportionately severe for samples.

Existing methods, designed with a holistic perspective, are ill-equipped to handle instance-level deviations arising from long-tailed forget distributions. Therefore, an instance-level unlearning method is in urgent need, and similar ideas have also been reflected in (Zhao et al., 2024).

To address this challenge, we propose *FaLW*, a plug-and-play, instance-wise dynamic loss reweighting method. *Our key innovation is to estimate the unlearning deviation for each sample by measuring its predictive probability against the distribution of probabilities for unseen data of the same class.* Based on this, we design a forgetting-aware dynamic weight. Furthermore, we introduce a balancing factor to modulate the sensitivity of these weights, making the adjustment more aggressive for tail-class samples. FaLW dynamically assesses the unlearning state of each instance and adjusts its unlearning intensity accordingly, effectively mitigating both heterogeneous and skewed unlearning deviations. In summary, our main contributions are as follows:

❶ *Brand-new Perspective:* This is the first work to investigate the long-tailed unlearning problem and formalize the phenomena of *Skewed* and *Heterogeneous Unlearning Deviation*, filling a key gap in the current field.

❷ *Plug-and-play Solution:* We propose to utilize the predicted probability distribution over unseen data to assess the instance-wise unlearning state during the process. Based on this, we introduce *FaLW*, a forgetting-aware dynamic loss reweighting that effectively mitigates the identified unlearning deviations in long-tailed scenarios.

❸ *Superior Effect:* Extensive experiments on multiple image classification benchmarks demonstrate that our method achieves state-of-the-art performance.

## 2 RELATED WORK

▶ **Machine Unlearning (MU).** The field of Machine Unlearning (MU) has emerged in response to the growing need to selectively remove the influence of specific data from previously trained models (Shaik et al., 2024; Xu et al., 2024; Neel et al., 2021; Sekhari et al., 2021). Methodologies for exact MU, which offer provable guarantees of data removal, have been thoroughly explored, especially for linear and convex models (Guo et al., 2019; Koh & Liang, 2017; Giordano et al., 2019). However, applying exact unlearning to deep learning models typically necessitates a complete retraining from

scratch (Bourtoule et al., 2021), a procedure that is computationally prohibitive and thus impractical for a majority of real-world applications. To enhance efficiency, an alternative approach, termed approximate MU (Xu et al., 2024; Thudi et al., 2022a), has gained prominence. A multitude of gradient-based approximate unlearning techniques (Golatkar et al., 2020; Warnecke et al., 2021; Jia et al., 2023; Graves et al., 2021; Thudi et al., 2022a; Chundawat et al., 2023; Cheng et al., 2024; Fan et al., 2024; Huang et al., 2024) have been introduced, which utilize specialized loss functions to guide the model in erasing information related to designated data, consequently mitigating privacy vulnerabilities like membership inference attacks (Hu et al., 2022; Li & Zhang, 2021; Song et al., 2019; Carlini et al., 2022) and data reconstruction attacks (Fredrikson et al., 2015; Balle et al., 2022). A significant challenge, akin to catastrophic forgetting in continual learning (McCloskey & Cohen, 1989; Wang et al., 2024; Ratcliff, 1990), arises when approximate methods focus excessively on the forget set, often causing a substantial decline in model performance on the remaining data. While various strategies (Tarun et al., 2023; Kurmanji et al., 2023; Fan et al., 2024; Huang et al., 2024), notably fine-tuning on the retain set, have been introduced to preserve model utility, we assert that a critical real-world condition has been overlooked. *Real-world unlearning data often follows long-tailed distributions, yet existing methods remain unexplored in long-tailed MU regimes.* Our FaLW fills this gap, enhancing long-tailed approximate unlearning performance.

▶ **Long-tailed Learning (LTL).** Existing LTL methods can be broadly classified into re-sampling and re-weighting, transfer learning and knowledge distillation, and multi-expert or modular designs. Specifically, re-sampling(Chawla et al., 2002; He et al., 2008) and re-weighting(Cui et al., 2019; Cao et al., 2019) techniques balance the data distribution or the learning process by adjusting sample probabilities or loss weights. Transfer learning(Yin et al., 2019; Liu et al., 2019) and knowledge distillation(Xiang et al., 2020; He et al., 2021) methods attempt to transfer knowledge from head classes to tail classes or leverage knowledge from large pre-trained models. However, the long-tailed problem in MU is distinct from traditional LTL. The distinction lies in the objective: while traditional methods aim to improve performance on tail classes, long-tailed unlearning must effectively remove information from an imbalanced set while simultaneously accounting for the unlearning performance changes that long-tailed data induce on both head and tail classes. Therefore, developing imbalance-aware methods specifically tailored for the machine unlearning paradigm is urgently needed. In this paper, motivated by our findings in the long-tailed unlearning scenario, we design a forgetting-aware approximate unlearning method suitable for this challenging setting.

## 3 PRELIMINARIES

▶ **Notation.** We consider a probability distribution $\mathbb{P}_{\mathcal{X}}$ over the input space $\mathcal{X}$ and a set of $C$ classes denoted by $\mathcal{Y} = \{1, 2, \ldots, C\}$. We focus on a multi-class classifier $\mathcal{F} : \mathcal{X} \to \mathcal{Y}$. For a given input $x \in \mathcal{X}$, the model's prediction function, $f(x)$, outputs a vector of predicted probabilities $p(\cdot|x) \in \mathbb{R}^C$ over the classes. The loss function for the model $\mathcal{F}$ is denoted by $\ell_{\mathcal{F}} : \mathcal{X} \times \mathcal{Y} \to \mathbb{R}_+$, which quantifies the discrepancy between the predicted probabilities and the ground-truth label $y$. In the supervised learning setting, we are given a dataset $\mathcal{D} = \{(x_i, y_i)\}_{i=1}^N$, consisting of $\mathcal{N}$ labeled samples where each input $x_i$ is drawn from $\mathbb{P}_{\mathcal{X}}$ and $y_i \in \mathcal{Y}$ is its corresponding label. The model $\mathcal{F}$, parameterized by $\boldsymbol{\theta}$, is trained on $\mathcal{D}$ by minimizing the empirical risk, often expressed as $\mathbb{E}_{(x,y) \in \mathcal{D}}[\ell_{\mathcal{F}}(x, y)]$. The resulting set of parameters from training is denoted by $\boldsymbol{\theta}_o$.

▶ **Machine Unlearning.** Let $\boldsymbol{\theta}_o$ denote the parameters of the initial model trained on the entire dataset $\mathcal{D}$. Given a subset $\mathcal{D}_f \subset \mathcal{D}$ to be forgotten, its complement, the retain set, is defined as $\mathcal{D}_r = \mathcal{D} \backslash \mathcal{D}_f$. The most straightforward unlearning method is retraining from scratch, which trains a model exclusively on the retain set $\mathcal{D}_r$ to obtain the parameters $\boldsymbol{\theta}_* = \arg\min_{\boldsymbol{\theta}} \mathbb{E}_{(x,y) \in \mathcal{D}_r}[\ell_{\mathcal{F}}(x, y)]$. Typically, *retraining* is here considered the gold standard for unlearning. However, *retraining* is often computationally prohibitive. A practical approach is approximate unlearning, which applies an algorithm $\mathcal{M}$ to update the original parameters $\boldsymbol{\theta}_o$ by removing the influence of $\mathcal{D}_f$. The resulting unlearned model parameters are denoted by $\boldsymbol{\theta}_u = \mathcal{M}(\boldsymbol{\theta}_o, [\mathcal{D}_f, \mathcal{D}_r])$. The objective is to ensure that the behavior of the unlearned model with parameters $\boldsymbol{\theta}_u$ closely approximates that of the gold-standard retrained model with parameters $\boldsymbol{\theta}_*$.

▶ **Long-tailed Settings.** Let $\mathcal{N}_k$ denote the number of samples for class $k$. A long-tailed distribution refers to a highly imbalanced data distribution. The sample distribution can often be modeled by a power law, such as $\mathcal{N}_k \propto k^{-\gamma}$, where the imbalance factor $\gamma$ controls the degree of class imbalance.

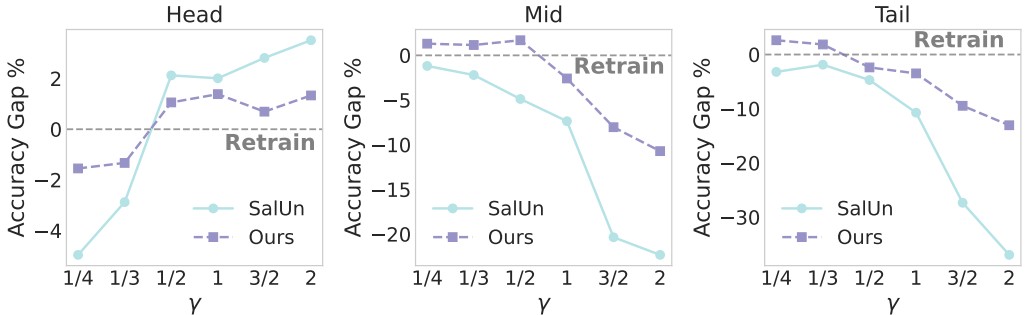

Figure 2: Under the setup of the **ResNet-18** architecture trained on **CIFAR-100** dataset with a **30%** unlearning ratio and a forget set distribution defined by $\mathcal{N}_k \propto \frac{1}{k^\gamma}$, this figure visualizes forgetting accuracy gaps between Salun, our method (Ours), and the Retrain baseline across head, mid, and tail classes within the forget set.

For notational convenience, we use $\mathcal{D}_{f,k}$ and $\mathcal{D}_{r,k}$ to denote the subsets of samples belonging to class $k$ within the forget set $\mathcal{D}_f$ and the retain set $\mathcal{D}_r$, respectively. Their cardinalities are given by $N_{f,k} = |\mathcal{D}_{f,k}|$ and $\mathcal{N}_{r,k} = |\mathcal{D}_{r,k}|$. We define the long-tailed unlearning problem as the scenario where the forget set $\mathcal{D}_f$ itself exhibits a long-tailed distribution, satisfying $\mathcal{N}_{f,1} \geq \mathcal{N}_{f,2} \geq \cdots \geq \mathcal{N}_{f,C}$ and $\mathcal{N}_{f,1} \gg \mathcal{N}_{f,C}$. Following standard practice in long-tailed learning, we partition the classes into head, medium, and tail groups based on the number of samples per class in the forget set, such that the total number of samples in each group satisfies $\mathcal{N}_{f,\text{head}} \gg \mathcal{N}_{f,\text{medium}} \gg \mathcal{N}_{f,\text{tail}}$.

## 4 MOTIVATION

**Definition 1** (Unlearning Deviation). *For any sample $(x_i, y_i) \in \mathcal{D}_f$ and a small positive threshold $\tau_i$, we categorize the unlearning outcome for $x_i$ by comparing the output probability of the approximate model $\boldsymbol{\theta}_u$ with that of the retrained model $\boldsymbol{\theta}_*$ on the ground-truth class $c = y_i$. The outcome is classified as:*

$$
\begin{cases}
❶ \textit{Under-forgetting}, & \textit{if } p_{\boldsymbol{\theta}_u}(c|x_i) > p_{\boldsymbol{\theta}_*}(c|x_i) + \tau_i \\
❷ \textit{Faithful-forgetting}, & \textit{if } |p_{\boldsymbol{\theta}_u}(c|x_i) - p_{\boldsymbol{\theta}_*}(c|x_i)| \leq \tau_i \\
❸ \textit{Over-forgetting}, & \textit{if } p_{\boldsymbol{\theta}_u}(c|x_i) < p_{\boldsymbol{\theta}_*}(c|x_i) - \tau_i
\end{cases}
\tag{1}
$$

Here, $p_{\boldsymbol{\theta}}(c|x_i)$ denotes the probability assigned to the true class $c$ for input $x_i$ by a model with parameters $\boldsymbol{\theta}$. The conditions of *under-forgetting* and *over-forgetting* are collectively termed ***Unlearning Deviation***.

**Formulation 1** (Gradient-based Approximate Unlearning). *Briefly, gradient-based approximate unlearning methods can be formally expressed in three parts:*

$$
\min_{\boldsymbol{\theta}_u} \ell_{unlearn}(\boldsymbol{\theta}_u) = \underbrace{\alpha \cdot \mathcal{L}(\mathcal{D}_f; \boldsymbol{\theta}_u)}_{❶ \textit{ Forgetting Execution}} + \underbrace{\beta \cdot \mathcal{L}(\mathcal{D}_r; \boldsymbol{\theta}_u)}_{❷ \textit{ Capacity Preservation}} + \underbrace{\lambda \cdot \mathcal{R}(\boldsymbol{\theta}_u, \boldsymbol{\theta}_o)}_{❸ \textit{ Parameter Constraints}}
\tag{2}
$$

In Formulation 1, ❶ *Forgetting Execution* involves computing the inverse gradient derived from $\mathcal{D}_f$, which is essential for implementing the model's forgetting process. ❷ *Capacity Preservation* leverages $\mathcal{D}_r$ to preserve the model's predictive accuracy on the remaining data. This is achieved by either constraining the inverse gradient (e.g., via projection methods) or introducing a forward gradient during optimization. ❸ *Parameter Constraints* term, $\mathcal{R}(\cdot)$, imposes constraints on the updated parameters $\boldsymbol{\theta}_u$, for instance, by penalizing large deviations from the original parameters $\boldsymbol{\theta}_o$ or by enforcing properties like L1-sparsity. The coefficients $\alpha, \beta$, and $\lambda$ are non-negative hyperparameters that balance the trade-offs between these competing terms.

**Proposition 1.** *When an approximate unlearning method has access only to the forget set $\mathcal{D}_f$, it is prone to severe over-forgetting. Formally, for an unlearned model $\boldsymbol{\theta}_u = \mathcal{M}(\boldsymbol{\theta}_o, \mathcal{D}_f)$, the condition for over-forgetting, $p_{\boldsymbol{\theta}_u}(c|x_i) < p_{\boldsymbol{\theta}_*}(c|x_i) - \tau_i$, is frequently met for samples $(x_i, c) \in \mathcal{D}_f$.*

The proof for Proposition 1 is deferred to **Appendix A.1**. This proposition underscores that successful unlearning does not require the model's confidence on a forgotten sample to be very low and even driven to zero. When the retain set $\mathcal{D}_r$ is unavailable, the extent of unlearning cannot be effectively calibrated, and the process lacks a natural stopping point, leading to over-forgetting. Consequently, most approximate unlearning methods leverage both $\mathcal{D}_f$ and $\mathcal{D}_r$, utilizing the forward gradient on $\mathcal{D}_r$ (the capacity preservation term) to counteract the aggressive parameter updates from the forgetting term. To further investigate this interplay, we conduct experiments on SalUn (Fan et al., 2024), which embodies all components of the objective in Formulation 1.

**Observation 1.** *As illustrated in **Figure 1(b)** and **Appendix B.1**, the predictive probabilities of the unlearned model $\boldsymbol{\theta}_u$ produced by SalUn are consistently lower than those of the retrained model $\boldsymbol{\theta}_*$ across various experimental settings. This indicates that despite leveraging the retain set for capacity preservation, the SalUn method still exhibits a discernible degree of over-forgetting.*

▶ **Random Sampling is more Balanced.** In many practical unlearning scenarios, the data designated for forgetting is inherently imbalanced. For instance, when a user requests the deletion of their account, a platform must ensure this user's data no longer influences its models (e.g., for recommendation or advertising). An individual's data, shaped by personal interests, is often highly skewed and concentrated within a specific subset of categories. However, prevailing research has largely simulated forget requests by randomly sampling from the training data. While random sampling could theoretically yield an imbalanced forget set, our empirical results indicate it consistently produces a class distribution that is far more balanced than a long-tailed one. Specifically, as shown in Figure 1(a), repeatedly sampling 20% of the CIFAR-100 dataset results in distributions that closely approximate a balanced one, starkly contrasting with the characteristics of a long-tailed distribution. This discrepancy highlights a critical gap: the behavior of approximate unlearning methods under the more realistic long-tailed forgetting scenario remains insufficiently explored.

▶ **When Unlearning Meets Imbalance.** To address this gap, we conduct extensive experiments on CIFAR-100, with details presented in Figure 2 and **Appendix B.2**. We construct long-tailed forget sets where the class distribution follows $\mathcal{N}_{f,k} \propto k^{-\gamma}$, with the exponent $\gamma$ controlling the severity of the imbalance. As we increase $\gamma$, we observe a critical trend: the unlearning outcome for head-class samples gradually shifts from over-forgetting to under-forgetting, while the under-forgetting for medium- and tail-class samples is exacerbated. We summarize these findings as follows:

**Observation 2.** *Under a long-tailed forget distribution, approximate unlearning methods exhibit two key essences:*

❶ ***Heterogeneous Unlearning Deviation:** The model demonstrates disparate unlearning patterns across classes, typically under-forgetting samples from head classes while over-forgetting those from tail classes.*

❷ ***Skewed Unlearning Deviation:** The magnitude of the unlearning deviation is disproportionately larger for tail-class data compared to head- and medium-class data.*

Existing approximate unlearning methods are predominantly designed with a holistic perspective, focusing on the aggregate forgetting effect. Consequently, there is a dearth of research on strategies to handle the differential unlearning behaviors that emerge across samples. This limitation becomes particularly acute when the forget set exhibits long-tailed characteristics. Therefore, an adaptive unlearning mechanism that can modulate the process at a finer, sample- or class-level granularity is urgently needed.

## 5 FaLW: A Forgetting-aware Loss Reweighting

Motivated by the challenges identified in Observation 2, we propose an instance-wise loss reweighting method. The overall architecture is depicted in Figure 3. This approach dynamically adjusts the loss weight for each sample based on its estimated unlearning deviation during the unlearning process, thereby steering the model towards a more faithful unlearning state. Specifically, we introduce a dynamic weight $w_i$ for each sample in $D_r$ and adapt the general objective from Formulation 1 as follows:

$$\min_{\boldsymbol{\theta}_u} \ell_{\text{unlearn}}(\boldsymbol{\theta}_u) = \alpha \sum_{(x_i, y_i) \in \mathcal{D}_f} w_i \cdot \mathcal{L}((x_i, y_i); \boldsymbol{\theta}_u) + \beta \cdot \mathcal{L}(\mathcal{D}_r; \boldsymbol{\theta}_u) + \lambda \cdot \mathcal{R}(\boldsymbol{\theta}_u, \boldsymbol{\theta}_o) \quad (3)$$

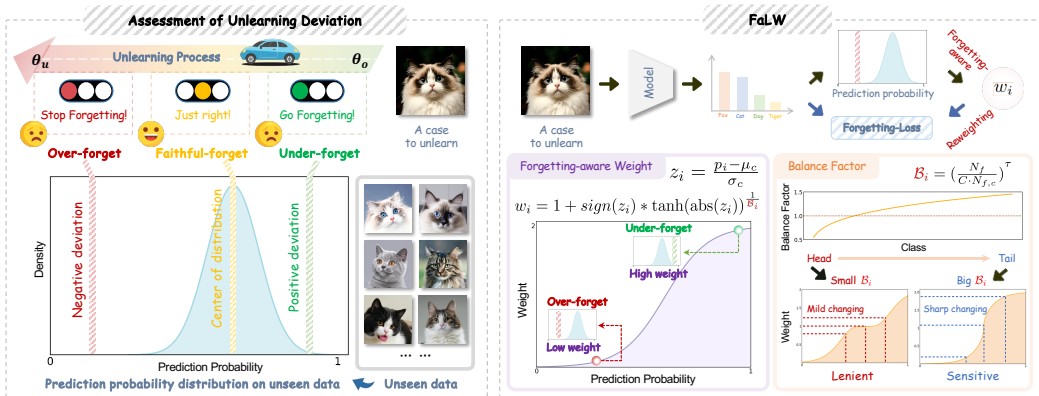

Figure 3: The FaLW framework. **Left**: Analyzing instance-level unlearning bias using the model's predictive probability on unseen data. **Right**: The top shows instance-level loss reweighting, and the bottom explains the forgetting-aware weight and balance factor.

## 5.1 ASSESSMENT OF UNLEARNING DEVIATION

*A Key Idea: The core principle of unlearning is to transition a model's state from having been trained on a sample to a state equivalent to never having seen it, briefly as from "seen" to "unseen".* Essentially, unlearning a sample aims to nullify its marginal contribution to the model's parameters, thereby reverting the model's predictive behavior on that sample to its naive state. To formalize this, let's consider unlearning a single sample $(x_i, y_i = c)$. The objective is to update the model parameters from the original $\boldsymbol{\theta}_o$ to the unlearned $\boldsymbol{\theta}_u$ such that the model's confidence on the true class $c$ approaches a specific target: $p_{\boldsymbol{\theta}_u}(c|x_i) \to p_{\boldsymbol{\theta}_*}(c|x_i)$. Here, $p_{\boldsymbol{\theta}_*}(c|x_i)$ represents the ideal predictive probability on $x_i$ from a model that has never been exposed to this sample. A way to get this ideal model, parameterized by $\boldsymbol{\theta}_*$, is retraining from scratch on the dataset with that specific sample removed: $\boldsymbol{\theta}_* = \arg\min_{\boldsymbol{\theta}} \mathbb{E}_{(x,y)\in\mathcal{D}\setminus\{(x_i,y_i)\}}[\ell_{\mathcal{F}}(x,y)]$.

**Proposition 2.** *All else being equal, a model trained on a dataset including a sample $(x_i, y_i = c)$ will exhibit higher confidence on that sample's true class compared to a model trained on the same dataset excluding it. Formally, let $\boldsymbol{\theta}_o$ be the parameters of a model trained on $\mathcal{D}$, and let $\boldsymbol{\theta}_*$ be the parameters of a model trained on $\mathcal{D} \setminus \{(x_i, y_i)\}$. Then, it generally holds that:*

$$p_{\boldsymbol{\theta}_o}(c|x_i) \geq p_{\boldsymbol{\theta}_*}(c|x_i) \tag{4}$$

*Remark.* The proof for Proposition 2 is provided in **Appendix A.2**. This proposition implies that as long as knowledge of a sample $x_i$ is retained, the model's predictive confidence will be inflated relative to the ideal unlearned state. If we assume for a moment that the target probability $p_{\boldsymbol{\theta}_*}(c|x_i)$ is accessible, Proposition 2 inspires a conceptual model for an ideal unlearning trajectory. Before the process begins, with parameters $\boldsymbol{\theta} = \boldsymbol{\theta}_o$, the model's confidence is high: $p_{\boldsymbol{\theta}}(c|x_i) \geq p_{\boldsymbol{\theta}_*}(c|x_i)$. As unlearning unfolds, $p_{\boldsymbol{\theta}}(c|x_i)$ should monotonically decrease, eventually converging to the target $p_{\boldsymbol{\theta}_*}(c|x_i)$. At this point, the sample is successfully forgotten, and the process should terminate to prevent over-forgetting. Therefore, during approximate unlearning, the discrepancy between the current confidence $p_{\boldsymbol{\theta}_u}(c|x_i)$ and the target $p_{\boldsymbol{\theta}_*}(c|x_i)$ can serve as a metric to gauge the extent of forgetting for sample $x_i$, provided the target is known.

▶ **Approximating** $\mathbf{p_{\boldsymbol{\theta}_*}(c|x_i)}$. In practice, obtaining the exact target probability $p_{\boldsymbol{\theta}_*}(c|x_i)$ is intractable during approximate unlearning. We therefore approximating the deterministic target into a target distribution. Specifically, we postulate that for a forgotten sample $(x_i, y_i = c)$, its ideal predictive probability $p_{\boldsymbol{\theta}_*}(c|x_i)$ should be indistinguishable from the probabilities of other samples of class $c$ that the model has never seen. Let $\mathbb{P}'_c$ be a distribution of unseen samples of class $c$. The ideal unlearning process should result in $p_{\boldsymbol{\theta}_*}(c|x_i)$ being a plausible draw from the distribution $\{p_{\boldsymbol{\theta}_*}(c|x')|x' \sim \mathbb{P}'_c\}$. As previously argued, unlearning aims to revert a sample's status from "seen" to "unseen". This distribution of probabilities on genuinely unseen data for class $c$ represents the desired endpoint. Crucially, while the target probability for any single sample is elusive, this target distribution can be estimated during the unlearning process.

▶ **Assessment of Unlearning Deviation.** To operationalize this concept, we model the distribution of predictive probabilities on unseen data for a given class $c$ with a Gaussian, i.e., $p_{\boldsymbol{\theta}}(c|x') \sim \mathcal{N}(\mu_c, \sigma_c^2)$ for any unseen sample $x' \sim \mathbb{P}_c'$. The Unlearning Deviation for a specific sample $(x_i, y_i = c)$ can then be dynamically assessed by measuring how its current probability $p_{\boldsymbol{\theta}}(c|x_i)$ deviates from this target distribution. This dynamic assessment guides the unlearning trajectory, as depicted in Figure 3 left. Initially, at $\boldsymbol{\theta} = \boldsymbol{\theta}_o$, the model's confidence $p_{\boldsymbol{\theta}}(c|x_i)$ is significantly higher than $\mu_c$, positioning it as a positive outlier of the distribution. This signifies a state of *under-forgetting*. As the unlearning process progresses, $p_{\boldsymbol{\theta}}(c|x_i)$ decreases. When its value falls within a high-density region of the distribution, we consider the sample faithfully forgotten, as its predictive signature is now typical of an unseen sample. At this stage, the unlearning pressure on this sample should be attenuated. If the process continues aggressively, $p_{\boldsymbol{\theta}}(c|x_i)$ may drop substantially below $\mu_c$, becoming a negative outlier and indicating a state of *over-forgetting*.

## 5.2 FORGETTING-AWARE WEIGHT

Based on the assessment method above, we introduce our Forgetting-Aware Weight $w_i$ for each sample $(x_i, y_i = c) \in \mathcal{D}_f$. The weight is formulated as:

$$w_i = 1 + \text{sign}(z_i) \cdot (\tanh(|z_i|))^{\frac{1}{\eta}}, \quad \text{with } z_i = \frac{p_i - \mu_c}{\sigma_c} \tag{5}$$

where $p_i = p_{\boldsymbol{\theta}}(c|x_i)$ is the model's current predictive probability on the sample. Here, $z_i$ is the standard z-score, quantifying how many standard deviations the current probability $p_i$ is from the mean $\mu_c$ of the target distribution for class $c$. The $\text{sign}(\cdot)$ function determines the direction of adjustment, while the hyperbolic tangent, $\tanh(\cdot)$, maps the magnitude of the deviation into the range $(-1, 1)$. The hyperparameter $\eta > 0$ acts as a temperature, controlling the sensitivity of the reweighting scheme; a larger $\eta$ leads to a more pronounced response to the deviation, causing $w_i$ to change more sharply. In practice, the class-wise $\mu_c$ and $\sigma_c$ are estimated by running the current model on a held-out validation set and calculating the mean and standard deviation of predictions for each class.

*Remark.* As depicted in the Figure 3 right, the reweighting scheme provides an intuitive, adaptive control mechanism. When a sample $x_i$ is over-forgotten, its confidence $p_i$ falls far below the mean $\mu_c$, resulting in a large negative z-score $z_i$. Consequently, $\text{sign}(z_i)$ becomes -1 and $w_i$ approaches zero ($1 - 1 = 0$), effectively halting the forgetting pressure on this sample. Conversely, for an under-forgotten sample, $p_i$ is a positive outlier, yielding a large positive $z_i$ and a weight $w_i$ that approaches 2 ($1 + 1 = 2$), thereby intensifying the unlearning process. This adaptive mechanism is particularly crucial for addressing the Heterogeneous Unlearning Deviation observed in long-tailed settings, as it holistically mitigates the deviation for all forget samples by preventing both over- and under-forgetting.

## 5.3 BALANCE FACTOR

To counteract the Skewed Unlearning Deviation, where tail-class samples suffer from more severe deviation, we introduce a class-aware balance factor $\mathcal{B}_i$ to make the reweighting scheme more sensitive for minority classes. We want the weight adjustment to be more aggressive for tail-class samples; for a given deviation, their weights should change more rapidly than those of head-class samples. To this end, we define the balance factor as:

$$\mathcal{B}_i = \left( \frac{N_f}{C \cdot N_{f,k}} \right)^{\tau} \tag{6}$$

where $c = y_i$, $\mathcal{N}_f = |\mathcal{D}_f|$ is the total number of samples in the forget set, $\mathcal{N}_{f,k}$ is the number of samples of class $c$ in the forget set, $C$ is the total number of classes, and $\tau$ is a non-negative hyperparameter. This factor is inversely proportional to the class frequency, yielding a larger value for tail-class samples. We then integrate this factor into our forgetting-aware weight by modulating the exponent:

$$w_i = 1 + \text{sign}(z_i) \cdot (\tanh(|z_i|))^{\frac{1}{\mathcal{B}_i}} \tag{7}$$

With this modification, for a tail-class sample with a large $\mathcal{B}_i$. This causes weight $w_i$ to react more sensitively to z-score $z_i$, thus enabling a more rapid response to its unlearning deviation. By

| Method | Tiny-Imagenet, Vgg-16 , $\gamma = 1$ | | | | | | CIFAR-10, Vgg-16 , $\gamma = 1/2$ | | | | | |
| | Random Forget (10%) | | | | | | Random Forget (40%) | | | | | |
| | FA | RA | TA | MIA | Avg. Gap | std | FA | RA | TA | MIA | Avg. Gap | std |
|---|---|---|---|---|---|---|---|---|---|---|---|---|
| Retrain | 25.63 (0.00) | 99.98 (0.00) | 48.83 (0.00) | 74.36 (0.00) | 0.00 | - | 65.94 (0.00) | 100.00 (0.00) | 78.62 (0.00) | 34.06 (0.00) | 0.00 | - |
| FT | 86.97 (61.33) | 99.98 (0.00) | 53.95 (5.12) | 13.03 (61.33) | 31.95 | 2.85 | 68.42 (2.48) | 96.51 (3.48) | 78.37 (0.25) | 31.58 (2.48) | 2.18 | 0.39 |
| RL | 18.50 (7.13) | 99.83 (0.15) | 48.51 (0.32) | 81.50 (7.14) | 3.69 | 0.94 | 72.37 (6.43) | 98.87 (1.13) | 81.15 (2.53) | 27.63 (6.43) | 4.13 | 0.99 |
| GA | 77.98 (52.34) | 93.73 (6.26) | 48.67 (0.16) | 22.02 (52.34) | 27.77 | 2.06 | 58.67 (7.27) | 84.28 (15.71) | 67.21 (11.41) | 41.33 (7.27) | 10.42 | 3.09 |
| IU | 96.29 (70.66) | 95.46 (4.52) | 55.71 (6.88) | 3.71 (70.65) | 38.18 | 0.18 | 98.84 (32.90) | 99.20 (0.80) | 90.47 (11.85) | 1.16 (32.90) | 19.61 | 0.50 |
| BE | 74.92 (49.29) | 92.61 (7.37) | 45.89 (2.94) | 25.08 (49.28) | 27.22 | 0.90 | 91.61 (25.67) | 98.26 (1.74) | 85.38 (6.76) | 8.39 (25.67) | 14.96 | 1.36 |
| BS | 72.60 (46.97) | 92.42 (7.57) | 46.25 (2.58) | 27.40 (46.96) | 26.02 | 0.83 | 82.61 (16.67) | 95.93 (4.06) | 80.83 (2.21) | 17.39 (16.67) | 9.90 | 1.25 |
| $l_1$-sparse | 96.33 (70.70) | 95.47 (4.52) | 55.19 (6.36) | 3.67 (70.69) | 38.07 | 0.77 | 82.58 (16.64) | 95.44 (4.55) | 82.95 (4.33) | 17.42 (16.64) | 10.54 | 1.71 |
| SFRon | 22.01 (3.62) | 97.10 (2.88) | 44.23 (4.60) | 77.98 (3.62) | 3.68 | 0.76 | 68.08 (2.14) | 97.10 (2.90) | 82.15 (3.53) | 30.91 (3.15) | 2.93 | 1.02 |
| SalUn | 22.68 (2.95) | 99.18 (0.80) | 46.11 (2.72) | 71.02 (3.34) | 2.45 | 0.41 | 63.84 (2.09) | 96.04 (3.96) | 77.03 (1.59) | 33.16 (0.91) | 2.14 | 0.15 |
| **FalW** | 25.70 (0.07) | 99.98 (0.00) | 49.11 (0.28) | 73.30 (1.06) | **0.35** | 0.20 | 65.64 (0.30) | 99.99 (0.00) | 78.01 (0.61) | 33.36 (0.70) | **0.40** | 0.19 |

Table 1: Results of comparative results with two setups: training **VGG-16** on **CIFAR-10** with **10%** forgetting rate and $\gamma = 1$; training **VGG-16** on **Tiny-ImageNet** with **40%** forgetting rate and $\gamma = 1/2$. ($\cdot$) indicates performance gaps between approximate unlearning methods and the Retrain method across metrics. "std" is the variance of Avg. Gap across multiple experiments.

incorporating this balance factor, our final reweighting scheme can simultaneously perceive and mitigate both Heterogeneous & Skewed Unlearning Deviations.

▶ **Conclusion.** Our method, FaLW, substantially mitigates the two types of unlearning deviation identified in Observation 2. For the *Heterogeneous Unlearning Deviation*, the forgetting-aware weight promotes the forgetting of under-forgotten samples by increasing their loss, while preventing further forgetting of over-forgotten samples by reducing their loss. For the *Skewed Unlearning Deviation*, the balance factor adjusts the sensitivity of this process, applying a more aggressive deviation correction for tail-class samples.

## 6 EXPERIMENTS

### 6.1 EXPERIMENTS SETTINGS

▶ **Datasets, Models, and Settings.** Our empirical evaluation is conducted on image classification unlearning tasks. We utilize ResNet-18 (He et al., 2016) and VGG-16 (Simonyan & Zisserman, 2014) as the primary model architectures. The experiments are performed on several standard benchmarks, including CIFAR-100 (Krizhevsky et al., 2009), CIFAR-10 (Krizhevsky et al., 2009), and Tiny-ImageNet (Le & Yang, 2015). We vary the size of the forget set from 10% to 50% of the total training data. To simulate imbalanced forget requests, we construct long-tailed forget sets where the class distribution follows a power law $\mathcal{N}_k \propto k^{-\gamma}$, where $\mathcal{N}_k$ is the number of samples in class $k$. We explore a wide range of imbalance factors by setting $\gamma \in \{0, 1/4, 1/3, 1/2, 1, 3/2, 2\}$, where $\gamma = 0$ corresponds to the balanced case(random sampling closer to).

▶ **Baselines and Evaluation.** We compare our method against 9 baselines, including four gradient-based methods: Fine-Tuning (FT) (Warnecke et al., 2021), Gradient Ascent (GA) (Thudi et al., 2022a), Random Label (RL) (Golatkar et al., 2020), SalUn (Fan et al., 2024) and SFRon (Huang et al., 2024); two boundary-based methods (Chen et al., 2023): Boundary Expanding (BE) and Boundary Shrinking (BS); one influence-function-based method: Influence Unlearning (IU) (Izzo et al., 2021); and one parameter-sparsification method: L1-Sparse (Jia et al., 2023). The detailed algorithmic implementation of our proposed FaLW, as employed in the experiments, is provided in **Appendix C.1**. To comprehensively evaluate performance, we use five primary metrics: ❶ **Forgetting Accuracy (FA)**, the accuracy of the unlearned model $\theta_u$ on the forget set $\mathcal{D}_f$; ❷ **Membership Inference Attack (MIA)**, a privacy measure evaluated on $\mathcal{D}_f$ to assess how effectively the model has forgotten the data; ❸ **Retain Accuracy (RA)**, the accuracy on the retain set $\mathcal{D}_r$, which measures the model's fidelity to the remaining data; ❹ **Test Accuracy (TA)**, the accuracy on the original test set, reflecting the model's overall generalization ability; ❺ **Avg. Gap**, the average of the above four metrics.

## 6.2 Experiments Results

▶ **Comparative Results.** We conducted extensive experiments across multiple datasets, forget ratios, and imbalance settings. Table 1 presents the performance of FaLW against baselines, with two setups: VGG-16 trained on CIFAR-10 (10% forgetting rate, $\gamma = 1$) and VGG-16 trained on Tiny-ImageNet (40% forgetting rate, $\gamma = 1/2$). As shown, FaLW achieves a significantly lower average performance gap (Avg. Gap) compared to all other approximate unlearning baselines. Furthermore, FaLW demonstrates superior performance across nearly all individual metrics (FA, RA, TA, and MIA). **Additional results** on Tiny-ImageNet and CIFAR-10, as well as under different forget ratios, different imbalance settings, and different model architectures, are available in **Appendix C.2**.

▶ **Results for Various Degrees of Imbalance.** To analyze the robustness under different imbalance scenarios, we conducted experiments on CIFAR-100 with a 30% forget ratio while varying the imbalance factor $\gamma$ from 0 (balanced distribution) to 2 (highly long-tailed). The results are summarized in Table 2. Across all degrees of imbalance, FaLW consistently achieves a lower Avg. Gap than SalUn, indicating that its performance more closely approximates the Retrain gold standard. Furthermore, we observe a telling trend in SalUn's behavior: at low imbalance levels, its Forgetting Accuracy (FA) is lower than Retrain's, suggesting over-forgetting. As the imbalance becomes more severe, SalUn's FA surpasses that of Retrain, indicating a shift towards under-forgetting. In contrast, FaLW maintains FAs much closer to the Retrain baseline regardless of the imbalance settings. This demonstrates that FaLW effectively mitigates the heterogeneous unlearning deviation that plagues existing methods.

|  | Method | FA | RA | TA | MIA | Avg. Gap |
|---|---|---|---|---|---|---|
| $\gamma = 0$ | Retrain | 59.11 | 99.95 | 59.74 | 40.89 | 0.00 |
|  | SaLUn | 56.93 | 99.96 | 60.60 | 44.06 | 1.55 |
|  | FaLW | 59.09 | 99.98 | 61.38 | 41.91 | **0.68** |
| $\gamma = \frac{1}{4}$ | Retrain | 57.13 | 99.96 | 59.24 | 42.87 | 0.00 |
|  | SaLUn | 55.19 | 99.96 | 59.98 | 39.41 | 1.54 |
|  | FaLW | 57.73 | 99.98 | 60.67 | 41.27 | **0.91** |
| $\gamma = \frac{1}{3}$ | Retrain | 56.29 | 99.97 | 59.33 | 43.71 | 0.00 |
|  | SaLUn | 54.00 | 99.98 | 59.65 | 40.18 | 1.54 |
|  | FaLW | 56.13 | 99.98 | 61.77 | 42.87 | **0.86** |
| $\gamma = \frac{1}{2}$ | Retrain | 50.36 | 99.95 | 57.71 | 49.64 | 0.00 |
|  | SaLUn | 48.47 | 99.96 | 58.66 | 51.53 | 1.19 |
|  | FaLW | 50.76 | 99.98 | 58.62 | 47.24 | **0.93** |
| $\gamma = 1$ | Retrain | 19.30 | 99.96 | 50.70 | 80.70 | 0.00 |
|  | SaLUn | 21.75 | 99.97 | 53.95 | 78.25 | 2.04 |
|  | FaLW | 19.69 | 99.97 | 52.63 | 79.31 | **0.93** |
| $\gamma = \frac{3}{2}$ | Retrain | 4.43 | 99.95 | 48.13 | 95.57 | 0.00 |
|  | SaLUn | 5.75 | 99.96 | 51.00 | 96.25 | 1.22 |
|  | FaLW | 4.64 | 99.87 | 50.47 | 96.36 | **0.85** |
| $\gamma = 2$ | Retrain | 1.82 | 99.94 | 47.12 | 98.17 | 0.00 |
|  | SaLUn | 4.79 | 99.96 | 50.32 | 95.20 | 2.29 |
|  | FaLW | 2.96 | 99.85 | 49.95 | 97.04 | **1.30** |

Table 2: Unlearning performance metrics of SalUn and FaLW with **ResNet-18** architecture trained on **CIFAR-100** by a **30%** unlearning ratio, under varying imbalance levels of the forget set controlled by $\gamma$ (larger $\gamma$ denotes greater class imbalance in the forget set).

▶ **Ablation Study on Balance Factor.** We conduct an ablation study to analyze the benefits of Balance Factor. The experiments are performed on CIFAR-100 with ResNet-18 and a 30% forget ratio, focusing on two highly skewed scenarios with IFs $\gamma = 1.5$ and $\gamma = 2.0$. In Table 3, the findings reveal a crucial trade-off: while the inclusion of the balancing factor can cause a minor decline in the forgetting accuracy for head-class data, it leads to a substantial improvement for tail-class data.

| $\gamma$ | Balance Factor | $\Delta$FA | $\Delta$Head FA | $\Delta$Mid FA | $\Delta$Tail FA |
|---|---|---|---|---|---|
| 1.5 | ✗ | 0.18 | 0.51 | -9.98 | -12.19 |
|  | ✔ | 0.20 | 0.69 | -8.04 | -9.46 |
| 2 | ✗ | 1.08 | 1.22 | -12.56 | -18.76 |
|  | ✔ | 1.14 | 1.33 | -10.71 | -13.04 |

Table 3: Forgetting Accuracy gaps ($\Delta$FA) vs. Retrain on **CIFAR-100** (**30%** unlearning, **ResNet-18**) with a power-law forget set ($\mathcal{N}_k \propto k^{-1.5}$ and $\mathcal{N}_k \propto k^{-2}$). Gaps are shown for the overall set ($\Delta$FA) and head/mid/tail classes.

This result confirms that the Balance Factor effectively fulfills its designed purpose—to specifically target and alleviate the skewed unlearning deviation.

▶ **Analysis of FaLW's Effectiveness.** Figure 2 validates FaLW's effectiveness in addressing unlearning deviation issues. Regarding *Heterogeneous Unlearning Deviation*, FaLW consistently re-

duces the performance gap whether the baseline over-forgets or under-forgets, correcting deviations in both directions. For *Skewed Unlearning Deviation*, it successfully mitigates the severe over-forgetting of tail-class samples that plagues the baseline in highly long-tailed settings.

▶ **Further Experiments.** We provide additional comparative experiments in **Appendix C.2**, t-SNE visualizations in **Appendix C.3**, a plug-and-play characteristic analysis in **Appendix F**, a hyperparameter sensitivity analysis for $\tau$ in **Appendix G**, and an analysis on the choice of normal distribution in **Appendix H**.

## 7 CONCLUSION

To address *Heterogeneous* and *Skewed Unlearning Deviation* in long-tailed unlearning, we propose **FaLW**, a plug-and-play dynamic loss reweighting that adaptively adjusts the unlearning intensity for each instance. Extensive experiments demonstrate that FaLW achieves SOTA performance.

## 8 REPRODUCIBILITY STATEMENT

To ensure the reproducibility of our work, we are providing the code for our experiments as part of the supplementary material. We also provide a description of relevant implementation details in Appendix C.1.

## 9 ETHICS STATEMENT

Our work adheres to the ICLR Code of Ethics. This research did not involve human subjects or animal experimentation. All datasets used in this study are publicly available and were handled in strict accordance with their usage terms to protect privacy. We have taken proactive measures to mitigate potential biases in our methodology and evaluation, ensuring no personally identifiable information was processed and no discriminatory outcomes were generated. We are committed to maintaining transparency and integrity throughout our research process.

## 10 ACKNOWLEDGEMENTS

The authors gratefully acknowledge the support from the National Natural Science Foundation of China (NSFC) under Grant Nos. 62402472, and 12227901. This work was also supported by the Natural Science Foundation of Jiangsu Province (No. BK20240461), the Project of Stable Support for Youth Team in Basic Research Field, CAS (No. YSBR-005), and the Academic Leaders Cultivation Program at USTC. The AI-driven experiments, simulations and model training were performed on the robotic AI-Scientist platform of Chinese Academy of Sciences.

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

# Appendix
# FaLW: A Forgetting-aware Loss Reweighting for Long-tailed Unlearning

This appendix is organized as follows:

- **Section A: Proofs.** This section provides the theoretical proofs for Proposition 1 and Proposition 2.

- **Section B: Observations.** This section presents additional experimental results for Observation 1 and Observation 2 to demonstrate the generality of our findings.

- **Section C: Experiments Complement.** This section details supplementary experiments, including Implementation Details, Additional Comparative Results, and a t-SNE Visualization Analysis.

- **Section D: Limitations.** This section discusses the limitations of the proposed FaLW method and outlines potential directions for future work.

- **Section E: LLM Usage Statement.** This section provides a statement on the use of Large Language Models (LLMs) in this research.

- **Section F: Plug-and-Play Characteristic Analysis of FaLW.** This section provides an analysis of the plug-and-play nature of the FaLW method, demonstrating its effectiveness when integrated with existing unlearning baselines.

- **Section G: Hyperparameter Analysis for $\tau$.** This section conducts a sensitivity analysis for the hyperparameter $\tau$ to evaluate its impact on the overall unlearning performance.

- **Section H: Clarification about use of the normal distribution.** This section justifies the choice of the Normal distribution for modeling predictive probabilities and provides a comparative experiment with a bounded Beta distribution variant.

## A   PROOF

### A.1   PROOF FOR PROPOSITION 1

**Proposition 1** (Proposed in paper). *When an approximate unlearning method has access only to the forget set $\mathcal{D}_f$, it is prone to severe over-forgetting. Formally, for an unlearned model $\boldsymbol{\theta}_u = \mathcal{M}(\boldsymbol{\theta}_o, \mathcal{D}_f)$, the condition for over-forgetting, $p_{\boldsymbol{\theta}_u}(c|x_i) < p_{\boldsymbol{\theta}_*}(c|x_i) - \tau_i$, is frequently met for samples $(x_i, c) \in \mathcal{D}_f$.*

*Proof.* When only $\mathcal{D}_f$ is available, approximate unlearning methods primarily update the model by applying gradients derived from this set, often designed to reverse the original training process. We illustrate this using the Random Labeling (RL) method as an example. In RL, for any sample $(x_i, y_i = c) \in \mathcal{D}_f$, its label is changed to an incorrect class $c' \neq c$. The model then performs gradient descent on a loss function computed with this incorrect label, which approximates applying an "unlearning" gradient. The standard cross-entropy loss for this relabeled sample is:

$$\mathcal{L}_{RL}(\boldsymbol{z}) = -\log(p_{c'}) \tag{8}$$

where $\boldsymbol{z} \in \mathbb{R}^C$ are the logit outputs of the model for sample $x_i$, and $p_j = e^{z_j} / \sum_{k=1}^{C} e^{z_k}$ is the softmax probability for class $j$. The gradients of this loss with respect to the logits of the true class $c$ and the incorrect class $c'$ are:

$$\frac{\partial \mathcal{L}_{RL}}{\partial z_j} = \begin{cases} p_c, & \text{if } j = c \\ p_{c'} - 1, & \text{if } j = c' \end{cases} \tag{9}$$

From Eq. 9, we can see that the unlearning process, driven by gradient descent on $\mathcal{L}_{RL}$, continuously pushes the probability of the true class, $p_c$, towards zero. The magnitude of this gradient, $p_c$, is larger when the model is more confident, accelerating the process. This dynamic, however, is misguided. Forgetting a sample is not equivalent to demanding that the model's confidence on it drops to zero. A model that has faithfully forgotten $x_i$ should still retain knowledge of class $c$ from the (now absent)

retain set, granting it a general capability to recognize class $c$. Even after unlearning, the model is expected to have some non-zero predictive capability for $x_i$ based on this general knowledge. Therefore, when only $\mathcal{D}_f$ is available, the unlearning process lacks a proper stopping condition and is prone to over-forgetting. This conclusion holds for other gradient-based methods as well, as they differ mainly in how the "unlearning" gradient is computed, not in the gradient update process. $\quad\square$

### A.2 PROOF FOR PROPOSITION 2

**Proposition 2** (Proposed in paper). *All else being equal, a model trained on a dataset including a sample $(x_i, y_i = c)$ will exhibit higher confidence on that sample's true class compared to a model trained on the same dataset excluding it. Formally, let $\boldsymbol{\theta}_o$ be the parameters of a model trained on $\mathcal{D}$, and let $\boldsymbol{\theta}_*$ be the parameters of a model trained on $\mathcal{D} \setminus \{(x_i, y_i)\}$. Then, it generally holds that:*

$$p_{\boldsymbol{\theta}_o}(c|x_i) \geq p_{\boldsymbol{\theta}_*}(c|x_i) \tag{10}$$

*Proof.* The proof is based on the principle of minimizing a loss function during model training. We use the standard cross-entropy loss (or negative log-likelihood) for classification. The cross-entropy loss for a single sample $(x_j, y_j)$ is given by:

$$\ell(\boldsymbol{\theta}; x_i, y_i) = -\log p_{\boldsymbol{\theta}}(y_i|x_i)$$

Let $\mathcal{D}_* = \mathcal{D} \setminus \{(x_i, y_i = c)\}$. The parameters $\boldsymbol{\theta}_*$ for the model trained on $\mathcal{D}_*$ are the solution to the following optimization problem:

$$\boldsymbol{\theta}_* = \arg\min_{\boldsymbol{\theta}} \mathcal{L}_*(\boldsymbol{\theta}) \quad \text{where} \quad \mathcal{L}_*(\boldsymbol{\theta}) = \sum_{(x_j, y_j) \in \mathcal{D}_*} L(\boldsymbol{\theta}; x_j, y_j)$$

The parameters $\boldsymbol{\theta}_o$ for the model trained on the full dataset $\mathcal{D} = \mathcal{D}_* \cup \{(x_i, y_i = c)\}$ are the solution to:

$$\boldsymbol{\theta}_o = \arg\min_{\boldsymbol{\theta}} \mathcal{L}_o(\boldsymbol{\theta})$$

where the total loss $\mathcal{L}_o(\boldsymbol{\theta})$ can be expressed as:

$$\mathcal{L}_o(\boldsymbol{\theta}) = \mathcal{L}_*(\boldsymbol{\theta}) + L(\boldsymbol{\theta}; x_i, y_i = c)$$

By the definition of $\boldsymbol{\theta}_o$ and $\boldsymbol{\theta}_*$ as the minimizers of their respective loss functions, we have the following two conditions:

1. Since $\boldsymbol{\theta}_o$ minimizes $\mathcal{L}_o(\boldsymbol{\theta})$, its value at $\boldsymbol{\theta}_o$ must be less than or equal to its value at any other point, including $\boldsymbol{\theta}_*$:
$$\mathcal{L}_o(\boldsymbol{\theta}_o) \leq \mathcal{L}_o(\boldsymbol{\theta}_*) \tag{11}$$

2. Similarly, since $\boldsymbol{\theta}_*$ minimizes $\mathcal{L}_*(\boldsymbol{\theta})$:
$$\mathcal{L}_*(\boldsymbol{\theta}_*) \leq \mathcal{L}_*(\boldsymbol{\theta}_o) \tag{12}$$

We begin with the first optimality condition equation 11 and substitute the definition of $\mathcal{L}_o$:

$$\mathcal{L}_*(\boldsymbol{\theta}_o) + L(\boldsymbol{\theta}_o; x_i, c) \leq \mathcal{L}_*(\boldsymbol{\theta}_*) + L(\boldsymbol{\theta}_*; x_i, c)$$

Rearranging the terms, we get:

$$L(\boldsymbol{\theta}_o; x_i, c) - L(\boldsymbol{\theta}_*; x_i, c) \leq \mathcal{L}_*(\boldsymbol{\theta}_*) - \mathcal{L}_*(\boldsymbol{\theta}_o)$$

From the second optimality condition equation 12, we know that the right-hand side, $\mathcal{L}_*(\boldsymbol{\theta}_*) - \mathcal{L}_*(\boldsymbol{\theta}_o)$, is less than or equal to zero. Therefore:

$$L(\boldsymbol{\theta}_o; x_i, c) - L(\boldsymbol{\theta}_*; x_i, c) \leq 0$$

Substituting the definition of the loss function $L$:

$$-\log p_{\boldsymbol{\theta}_o}(c|x_i) \leq -\log p_{\boldsymbol{\theta}_*}(c|x_i)$$

Since the logarithm function is monotonically increasing, this inequality holds if and only if:

$$p_{\boldsymbol{\theta}_o}(c|x_i) \geq p_{\boldsymbol{\theta}_*}(c|x_i)$$

$\quad\square$

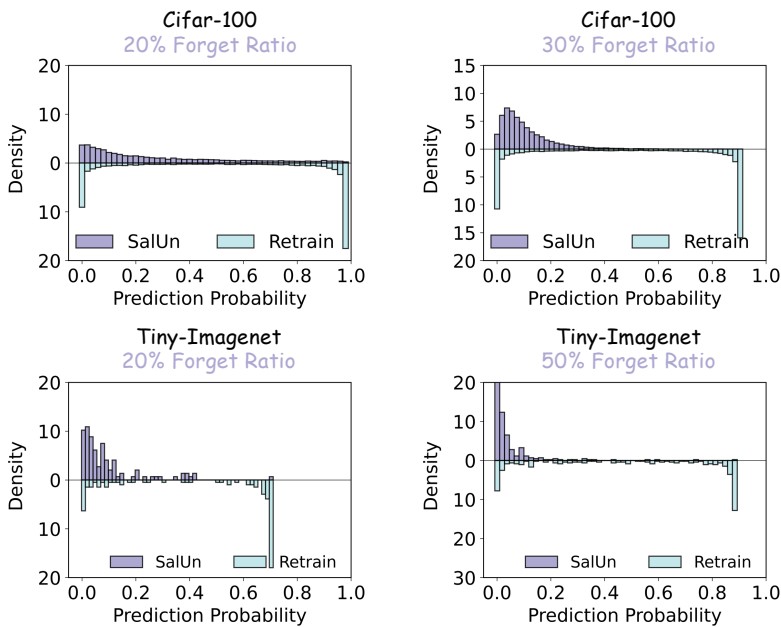

Figure 4: Comparative results of predicted probability distributions on randomly sampled forgotten data between the Salun and Retrain methods under multiple settings.

# B  OBSERVATION

## B.1  EXPERIMENTS FOR OBSERVATION 1

**Observation 1** (Proposed in paper). *The predictive probabilities of the unlearned model $\theta_u$ produced by SalUn are consistently lower than those of the retrained model $\theta_*$ across various experimental settings. This indicates that despite leveraging the retain set for capacity preservation, the SalUn method still exhibits a discernible degree of over-forgetting.*

Further results, presented in Figure 4, provide additional evidence of baseline limitations. It is observed that across multiple settings, the predictive probabilities of SalUn on the forget set are consistently and significantly lower than those of the Retrain. This demonstrates that even when leveraging the retain set and applying parameter constraints, the SalUn method is still prone to severe over-forgetting. Overall, the experimental results are consistent with our proposed Observation 1.

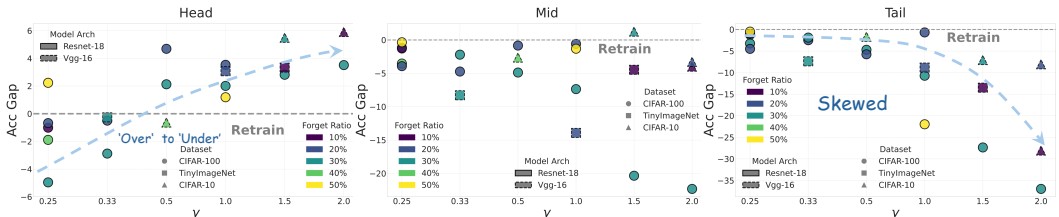

Figure 5: Performance discrepancies between the SalUn method and Retrain on the head, middle, and tail segments of forgotten data under different long-tailed configurations across various experimental settings.

## B.2  EXPERIMENTS FOR OBSERVATION 2

**Observation 2** (Proposed in paper). *Under a long-tailed forget distribution, approximate unlearning methods exhibit two key essences:*

❶ *Heterogeneous Unlearning Deviation:* *The model demonstrates disparate unlearning patterns across classes, typically under-forgetting samples from head classes while over-forgetting those from tail classes.*

❷ *Skewed Unlearning Deviation:* *The magnitude of the unlearning deviation is disproportionately larger for tail-class data compared to head- and medium-class data.*

We conducted further experiments under various imbalance levels and other settings, with detailed results shown in Figure 5. A consistent pattern emerges: as the degree of imbalance increases, the unlearning behavior for head-class data generally shifts from over-forgetting to under-forgetting, corroborating our first point in Observation 2. Conversely, for tail-class data, the unlearning performance deteriorates further as the imbalance becomes more pronounced, leading to more severe over-forgetting. This confirms our second point in Observation 2. Thus, Observation 2 is in line with the results of a large number of experiments.

---

**Algorithm 1** The FaLW Algorithm (RL-based version)

---

**Require:** Original model parameters $\theta_o$; Forget set $\mathcal{D}_f$; Retain set $\mathcal{D}_r$; Validation set $\mathcal{D}_{val}$.
**Require:** Hyperparameters: learning rate $\eta_{lr}$, exponents $\tau, \eta$, number of epochs $E$.
    $\theta \leftarrow \theta_o$
    Construct relabeled forget set $\mathcal{D}'_f \leftarrow \{(x_i, y'_i) | (x_i, y_i) \in \mathcal{D}_f, y'_i \neq y_i\}$
    Construct training set for unlearning $\mathcal{D}_{unlearn} \leftarrow \mathcal{D}'_f \cup \mathcal{D}_r$
    Pre-compute class-wise balancing factor $\mathcal{B}_c$ for each class $c$ using Eq. (6)
    **for** epoch $\leftarrow 1 \ldots E$ **do**
        **for** each batch $b \subset \mathcal{D}_{unlearn}$ **do**
            Estimate target distribution parameters $\mu_c, \sigma_c$ for each class $c$ from predictions on $\mathcal{D}_{val}$
            Partition the batch $b$ into forget part $b_f$ and retain part $b_r$
            Calculate retain loss $\{\mathcal{L}_{r,i}\} \leftarrow \text{loss}(b_r; \theta)$
            Calculate forget loss $\{\mathcal{L}_{f,i}\}$ and probabilities $\{p_i\}$ on $b_f$
            Calculate weights $\{w_{f,i}\}$ for samples in $b_f$ using $\mu_c, \sigma_c, \{p_i\}, \mathcal{B}_c$ via Eq. (7)
            $\mathcal{L}_{total} \leftarrow \text{mean}(\{w_{f,i} \cdot \mathcal{L}_{f,i}\}, \{\mathcal{L}_{r,i}\})$         ▷ Combine losses
            $\theta \leftarrow \theta - \eta_{lr} \nabla_\theta \mathcal{L}_{total}$         ▷ Update parameters via SGD
        **end for**
    **end for**
    **return** $\theta$

---

## C  EXPERIMENTS COMPLEMENT

### C.1  IMPLEMENTATION DETAILS

For all methods, we use an SGD optimizer with a momentum of 0.9, a weight decay of 5e-4, and a batch size of 512. For the Retrain, we train for 150 epochs with an initial learning rate of 0.01, which is decayed by a factor of 10 at epochs 90 and 120. For all baselines, we unlearn for 10 to 20 epochs, and the learning rate is tuned within the range of [0.0005, 0.02], with generally lower rates used for Tiny-ImageNet and higher rates for CIFAR-10 and CIFAR-100. For SalUn (Fan et al., 2024), we use the official implementation's default setting, which prunes the top 50% of parameters based on gradient magnitude. Similarly, for SFRon (Huang et al., 2024), we reproduced our results following the official code and hyperparameter settings provided in its original publication. The hyperparameter $\tau$ for our balancing factor in FaLW is tuned within the range [0.1, 0.2]. The pseudo-code for FaLW is provided in Algorithm 1, and our source code is included in the supplementary materials.

### C.2  MORE COMPARATIVE RESULTS.

We conducted a comprehensive evaluation of FaLW against baselines on CIFAR-100, CIFAR-10, and Tiny-ImageNet, performing a multi-faceted comparison across various forget ratios, imbalance levels, and model architectures. Table 4 presents the results for the ResNet-18 architecture on CIFAR-100 across multiple settings. The results for the VGG-16 architecture on CIFAR-10 and

Tiny-ImageNet are detailed in Table 6 and Table 5, respectively. The findings consistently show that our method, FaLW, outperforms the baselines in all evaluated settings, reducing the Avg. Gap by up to 3%.

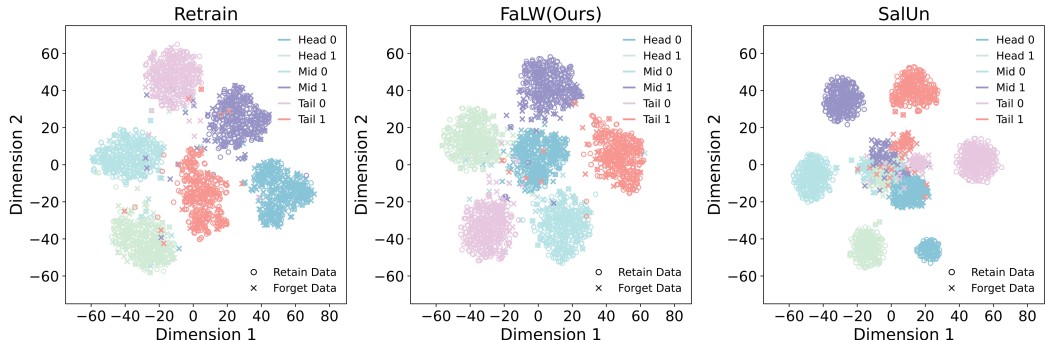

Figure 6: Experimental setup: **20%** of the **Cifar100** dataset is randomly selected as the forgetting set, following a data distribution of $\mathcal{N}_k \propto k^{-0.25}$. Under this setting, we select 2 head classes, 2 middle classes, and 2 tail classes, and visualize their representations via t-SNE across three methods: Retrain, SalUn, and FaLW. Each '$\bigcirc$' represents a sample in $\mathcal{D}_r$, while each '$\times$' denote a sample from $\mathcal{D}_f$.

### C.3 t-SNE VISUALIZATION ANALYSIS

To analyze the characteristics of forgotten data in the embedding space after approximate unlearning, we use t-SNE to visualize the representation. The experiment is set with ResNet-18 trained on CIFAR-100 by a 20% forget set following a distribution of $\mathcal{N}_k \propto k^{-0.25}$. We visualize samples from two heads, two medians, and two tails, with the results presented in Figure 6. It is evident that the feature distribution of FaLW closely resembles that of the gold-standard Retrain model. Specifically, for the forgotten samples, FaLW, much like Retrain, does not aggressively push their representations away from their original class clusters, thus achieving faithful unlearning. In stark contrast, SalUn displaces the forgotten samples far from the Retrain data clusters, exhibiting severe over-forgetting.

## D LIMITATIONS

While our proposed FaLW method demonstrates notable success in addressing and correcting deviations in machine unlearning, we acknowledge a primary limitation: its reliance on auxiliary data to evaluate the model's state during the forgetting process. This dependency on external data may limit the applicability of FaLW in resource-constrained or data-sensitive environments. Addressing this limitation constitutes a crucial direction for our future work.

## E LLM USAGE STATEMENT

In the preparation of this manuscript, we utilized a large language model (LLM) as a general-purpose writing assistant. The role of the LLM was strictly limited to improving the language and readability of our paper. This included tasks such as correcting grammatical errors, refining sentence structure for clarity, and enhancing overall prose. We confirm that the LLM did not contribute to the core research ideas, experimental design, data analysis, or the generation of any substantive content. All intellectual contributions, including the concepts, methodology, and conclusions presented in this paper, are solely the work of the human authors.

## F PLUG-AND-PLAY CHARACTERISTIC ANALYSIS OF FALW

As mentioned in the main text, FaLW is designed as a plug-and-play method. It is not a standalone unlearning algorithm but rather a forgetting-aware loss reweighting strategy intended for integration

with existing methods. To validate the efficacy of FaLW as a plug-and-play universal module, we integrated it into four baseline methods: GA (Thudi et al., 2022a), RL (Golatkar et al., 2020), SalUn (Fan et al., 2024), and SFRon (Huang et al., 2024). In terms of implementation, since SFRon also proposed a weighting term, we replaced its original dynamic weighting method with our FaLW's dynamic weighting. For the other three methods, we directly applied our weighting to the loss of the forget set samples. We conducted experiments on the CIFAR-100 dataset with the ResNet-18 architecture, a 20% forget rate, and $\gamma = 1/4$. The performance comparison before and after integrating FaLW is presented in Table 7. As the results show, the introduction of FaLW brings unlearning performance improvements to all baseline methods.

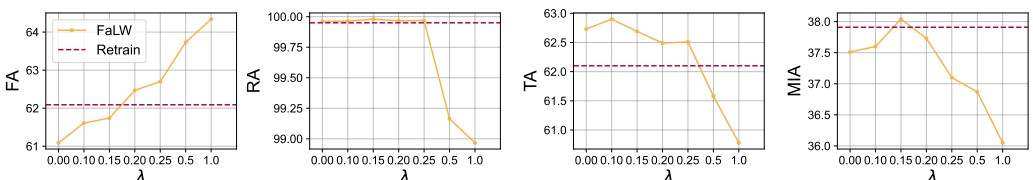

Figure 7: Hyperparameter sensitivity analysis for $\tau$ on CIFAR-100 (ResNet-18, $\gamma = 1/4$, 20% forget rate). The dashed red line represents the "Retrain" baseline performance for each metric.

## G  HYPERPARAMETER ANALYSIS FOR $\tau$

We conduct a sensitivity analysis for the hyperparameter $\tau$, which controls the exponent of the balance factor. The analysis is performed on the CIFAR-100 dataset using a ResNet-18 model, with a 20% forget rate and an imbalance ratio of $\gamma = 1/4$. We vary $\tau$ within the range [0, 1.0]. The results are illustrated in Figure 7. The performance across all metrics (FA, RA, TA, and MIA) is strongest and most stable when $\tau$ is set within the [0.1, 0.2] range. As $\tau$ increases beyond this optimal range, the model's deviation correction becomes overly sensitive, leading to a degradation in unlearning performance.

## H  CLARIFICATION ABOUT USE OF THE NORMAL DISTRIBUTION

Our initial choice of the Normal distribution was driven by practicality and robustness. We employ it not to perfectly fit the data density, but to derive a stable, efficient metric for "deviation" (z-score). In dynamic optimization scenarios, the Gaussian assumption offers computational simplicity and stable parameter estimation ($\mu$, $\sigma$), which helps avoid gradient instability.

To evaluate the impact of using a distribution with a matching support of $[0, 1]$, we implemented a variant named **FaLW-beta**. In this variant, we replace the Gaussian model with a Beta distribution. We estimate the parameters $\alpha_c$ and $\beta_c$ via Maximum Likelihood Estimation (MLE) on the validation set and adapt our dynamic weight formulation using the Beta Cumulative Distribution Function (CDF), $F(p_i) = \text{BetaCDF}(p_i | \alpha_c, \beta_c)$. We then map the probability to a deviation score $s_i \in [-1, 1]$:

$$w_i = 1 + \text{sign}(s_i) \cdot (|s_i|)^{\frac{1}{B_i}}, \quad \text{where } s_i = 2 \cdot (F(p_i) - 0.5) \tag{13}$$

We compared the FaLW-normal and FaLW-beta variants on the CIFAR-100 and Tiny-ImageNet datasets. The results are summarized in Table 8. While FaLW-beta yields reasonable performance, it consistently underperforms FaLW-normal, exhibiting a higher Average Gap. Empirically, we observed that the flexibility of the Beta distribution leads to high sensitivity and erratic weight fluctuations during the unlearning process, likely due to estimation errors in $\alpha_c$ and $\beta_c$. In contrast, the Normal distribution acts as a robust proxy, providing smoother gradients and greater stability "out of the box." Consequently, we retain the Normal distribution in our primary method for its robustness.

Table 4: Results on the **CIFAR-100** dataset, with the forget set following a distribution of $\mathcal{N}_k \propto \frac{1}{k^{0.25}}$. (·) indicates performance gaps between approximate unlearning methods and the Retrain method across metrics.

| Method | CIFAR-100, ResNet-18 , $\gamma = 1/4$ | | | | |
|---|---|---|---|---|---|
| | Random Forget (10%) | | | | |
| | FA | RA | TA | MIA | Avg. Gap |
| Retrain | 64.82 (0.00) | 99.95 (0.00) | 64.24 (0.00) | 35.18 (0.00) | 0.00 |
| FT | 96.27 (31.44) | 99.92 (0.03) | 70.30 (6.06) | 3.73 (31.44) | 17.24 |
| RL | 31.07 (33.76) | 97.52 (2.42) | 62.22 (2.02) | 48.93 (13.76) | 12.99 |
| GA | 95.22 (30.40) | 96.66 (3.29) | 68.83 (4.59) | 4.78 (30.40) | 17.17 |
| IU | 96.87 (32.05) | 96.93 (3.02) | 70.49 (6.25) | 3.13 (32.04) | 18.34 |
| BE | 96.71 (31.89) | 96.81 (3.13) | 69.61 (5.37) | 3.29 (31.89) | 18.07 |
| BS | 96.53 (31.71) | 96.60 (3.35) | 68.88 (4.64) | 3.47 (31.71) | 17.85 |
| $l_1$-sparse | 91.24 (26.42) | 91.38 (8.57) | 64.14 (0.10) | 8.76 (26.42) | 15.38 |
| SFRon | 62.69 (2.13) | 99.98 (0.03) | 66.70 (2.46) | 38.31 (3.13) | 1.94 |
| SalUn | 65.80 (0.98) | 99.95 (0.00) | 63.39 (0.85) | 32.20 (2.98) | 1.20 |
| **FalW** | 65.12 (0.30) | 99.98 (0.03) | 65.86 (1.62) | 33.58 (1.60) | **0.89** |

| Method | CIFAR-100, ResNet-18 , $\gamma = 1/4$ | | | | |
|---|---|---|---|---|---|
| | Random Forget (20%) | | | | |
| | FA | RA | TA | MIA | Avg. Gap |
| Retrain | 62.09 (0.00) | 99.95 (0.00) | 62.10 (0.00) | 37.91 (0.00) | 0.00 |
| FT | 96.61 (34.52) | 99.93 (0.02) | 69.98 (7.88) | 3.39 (34.52) | 19.24 |
| RL | 47.49 (14.60) | 96.99 (2.96) | 61.46 (0.64) | 42.51 (4.60) | 5.70 |
| GA | 95.22 (33.13) | 96.66 (3.30) | 68.83 (6.73) | 4.78 (33.13) | 19.07 |
| IU | 97.02 (34.93) | 96.91 (3.05) | 70.57 (8.47) | 2.98 (34.93) | 20.35 |
| BE | 95.41 (33.32) | 96.06 (3.89) | 65.83 (3.73) | 4.59 (33.32) | 18.57 |
| BS | 87.04 (24.95) | 87.67 (12.29) | 58.13 (3.97) | 12.96 (24.96) | 16.54 |
| $l_1$-sparse | 94.02 (31.93) | 93.88 (6.07) | 65.51 (3.41) | 5.98 (31.93) | 18.34 |
| SFRon | 65.30 (3.21) | 99.98 (0.03) | 66.42 (4.32) | 33.69 (4.22) | 2.95 |
| SalUn | 59.71 (2.38) | 96.92 (3.03) | 60.11 (1.99) | 40.29 (2.38) | 2.44 |
| **FalW** | 61.74 (0.34) | 99.98 (0.02) | 62.69 (0.59) | 38.26 (0.34) | **0.33** |

| Method | CIFAR-100, ResNet-18 , $\gamma = 1/4$ | | | | |
|---|---|---|---|---|---|
| | Random Forget (30%) | | | | |
| | FA | RA | TA | MIA | Avg. Gap |
| Retrain | 57.13 (0.00) | 99.96 (0.00) | 59.24 (0.00) | 42.87 (0.00) | 0.00 |
| FT | 96.32 (39.19) | 99.95 (0.01) | 69.80 (10.56) | 3.68 (39.19) | 22.23 |
| RL | 44.24 (12.90) | 96.34 (3.62) | 58.08 (1.16) | 65.76 (22.90) | 10.14 |
| GA | 85.51 (28.38) | 89.24 (10.72) | 61.40 (2.16) | 14.49 (28.38) | 17.41 |
| IU | 97.02 (39.88) | 96.89 (3.07) | 70.41 (11.17) | 2.98 (39.88) | 23.50 |
| BE | 75.60 (18.47) | 79.53 (20.43) | 48.09 (11.15) | 24.40 (18.47) | 17.13 |
| BS | 56.79 (0.34) | 58.94 (41.01) | 38.46 (20.78) | 43.21 (0.34) | 15.62 |
| $l_1$-sparse | 95.64 (38.50) | 95.38 (4.58) | 66.65 (7.41) | 4.36 (38.50) | 22.25 |
| SFRon | 54.46 (2.67) | 99.97 (0.01) | 60.72 (1.48) | 45.54 (2.67) | 1.71 |
| SalUn | 55.19 (1.95) | 99.96 (0.00) | 59.98 (0.74) | 39.41 (3.46) | 1.54 |
| **FalW** | 57.73 (0.60) | 99.98 (0.02) | 60.67 (1.43) | 41.27 (1.60) | **0.91** |

| Method | CIFAR-100, ResNet-18 , $\gamma = 1/4$ | | | | |
|---|---|---|---|---|---|
| | Random Forget (40%) | | | | |
| | FA | RA | TA | MIA | Avg. Gap |
| Retrain | 53.01 (0.00) | 99.95 (0.00) | 56.41 (0.00) | 46.99 (0.00) | 0.00 |
| FT | 96.24 (43.23) | 99.96 (0.01) | 69.46 (13.05) | 3.76 (43.23) | 24.88 |
| RL | 33.86 (19.15) | 95.17 (4.78) | 54.51 (1.90) | 66.14 (19.16) | 11.25 |
| GA | 96.88 (43.87) | 96.91 (3.04) | 70.44 (14.03) | 3.12 (43.87) | 26.20 |
| IU | 96.96 (43.95) | 96.91 (3.04) | 70.27 (13.86) | 3.04 (43.95) | 26.20 |
| BE | 96.92 (43.91) | 96.85 (3.10) | 70.33 (13.92) | 3.08 (43.91) | 26.21 |
| BS | 96.92 (43.91) | 96.88 (3.07) | 70.23 (13.82) | 3.08 (43.91) | 26.18 |
| $l_1$-sparse | 96.26 (43.24) | 96.23 (3.72) | 67.75 (11.34) | 3.74 (43.24) | 25.39 |
| SFRon | 51.33 (1.68) | 99.98 (0.03) | 58.59 (2.18) | 48.67 (1.68) | 1.39 |
| SalUn | 51.27 (1.74) | 99.97 (0.02) | 57.89 (1.48) | 49.73 (2.74) | 1.49 |
| **FalW** | 53.32 (0.31) | 99.98 (0.03) | 57.71 (1.30) | 45.68 (1.31) | **0.74** |

| Method | CIFAR-100, ResNet-18 , $\gamma = 1/4$ | | | | |
|---|---|---|---|---|---|
| | Random Forget (50%) | | | | |
| | FA | RA | TA | MIA | Avg. Gap |
| Retrain | 48.44 (0.00) | 99.94 (0.00) | 51.46 (0.00) | 51.56 (0.00) | 0.00 |
| FT | 95.79 (47.35) | 99.96 (0.02) | 69.15 (17.69) | 5.78 (45.78) | 27.71 |
| RL | 67.78 (19.34) | 96.30 (3.64) | 54.35 (2.89) | 28.41 (23.15) | 12.26 |
| GA | 96.56 (48.12) | 96.86 (3.08) | 69.10 (17.64) | 3.44 (48.12) | 29.24 |
| IU | 96.98 (48.54) | 96.88 (3.07) | 70.26 (18.80) | 3.02 (48.54) | 29.74 |
| BE | 96.90 (48.46) | 96.93 (3.01) | 70.11 (18.65) | 3.07 (48.49) | 29.65 |
| BS | 96.92 (48.48) | 96.88 (3.06) | 70.30 (18.84) | 3.08 (48.48) | 29.71 |
| $l_1$-sparse | 75.32 (26.88) | 88.76 (11.18) | 61.54 (10.08) | 24.68 (26.89) | 18.76 |
| SFRon | 46.30 (2.14) | 99.98 (0.04) | 54.78 (3.32) | 53.70 (2.14) | 1.91 |
| SalUn | 52.59 (4.15) | 96.12 (3.83) | 54.71 (3.25) | 48.15 (3.41) | 3.66 |
| **FalW** | 48.06 (0.38) | 99.98 (0.04) | 52.05 (0.59) | 52.94 (1.38) | **0.60** |

Table 5: Results on the **Tiny-Imagenet** dataset. (·) indicates performance gaps between approximate unlearning methods and the Retrain method across metrics.

| Method | Tiny-Imagenet, Vgg-16 , $\gamma = 0$ | | | | |
|---|---|---|---|---|---|
| | Random Forget (20%) | | | | |
| | FA | RA | TA | MIA | Avg. Gap |
| Retrain | 48.87 (0.00) | 99.98 (0.00) | 48.51 (0.00) | 51.13 (0.00) | 0.00 |
| FT | 95.62 (46.76) | 99.98 (0.00) | 55.27 (6.76) | 4.38 (46.75) | 25.07 |
| RL | 32.47 (16.40) | 98.25 (1.73) | 45.37 (3.14) | 67.53 (16.40) | 9.42 |
| GA | 94.61 (45.74) | 95.41 (4.57) | 54.27 (5.76) | 5.39 (45.73) | 25.45 |
| IU | 95.64 (46.77) | 95.56 (4.43) | 55.55 (7.04) | 4.36 (46.77) | 26.25 |
| BE | 94.41 (45.54) | 94.27 (5.71) | 49.87 (1.36) | 5.59 (45.54) | 24.54 |
| BS | 92.26 (43.39) | 92.45 (7.54) | 48.25 (0.26) | 7.74 (43.39) | 23.64 |
| $l_1$-sparse | 95.68 (46.82) | 95.53 (4.45) | 55.47 (6.96) | 4.32 (46.81) | 26.26 |
| SFRon | 50.23 (1.36) | 99.98 (0.00) | 50.11 (1.60) | 50.77 (0.36) | 0.83 |
| SalUn | 42.74 (6.13) | 99.13 (0.85) | 46.59 (1.92) | 56.26 (5.13) | 3.51 |
| **FalW** | 48.60 (0.27) | 99.98 (0.00) | 50.67 (2.16) | 50.40 (0.73) | **0.79** |

| Method | Tiny-Imagenet, Vgg-16 , $\gamma = 0$ | | | | |
|---|---|---|---|---|---|
| | Random Forget (50%) | | | | |
| | FA | RA | TA | MIA | Avg. Gap |
| Retrain | 40.60 (0.00) | 99.99 (0.00) | 40.63 (0.00) | 59.40 (0.00) | 0.00 |
| FT | 95.59 (54.99) | 99.99 (0.00) | 54.71 (14.08) | 4.41 (54.99) | 31.02 |
| RL | 32.16 (8.44) | 96.91 (3.08) | 37.33 (3.30) | 67.84 (8.44) | 5.82 |
| GA | 95.58 (54.97) | 95.49 (4.51) | 55.45 (14.82) | 4.42 (54.97) | 32.32 |
| IU | 95.65 (55.05) | 95.50 (4.50) | 55.65 (15.02) | 4.34 (55.05) | 32.41 |
| BE | 95.64 (55.04) | 95.47 (4.52) | 55.29 (14.66) | 4.36 (55.04) | 32.32 |
| BS | 95.66 (55.05) | 95.47 (4.53) | 55.03 (14.40) | 4.34 (55.05) | 32.26 |
| $l_1$-sparse | 95.68 (55.08) | 95.48 (4.52) | 55.45 (14.82) | 4.32 (55.08) | 32.37 |
| SFRon | 38.89 (1.71) | 99.99 (0.00) | 40.45 (0.18) | 61.10 (1.70) | 0.90 |
| SalUn | 38.49 (2.11) | 97.21 (2.78) | 36.23 (4.40) | 61.51 (2.11) | 2.85 |
| **FalW** | 41.76 (1.16) | 99.99 (0.00) | 40.53 (0.10) | 57.24 (2.16) | **0.85** |

| Method | Tiny-Imagenet, Vgg-16 , $\gamma = 1/3$ | | | | |
|---|---|---|---|---|---|
| | Random Forget (30%) | | | | |
| | FA | RA | TA | MIA | Avg. Gap |
| Retrain | 44.57 (0.00) | 99.98 (0.00) | 45.61 (0.00) | 55.43 (0.00) | 0.00 |
| FT | 90.44 (45.87) | 99.74 (0.25) | 51.19 (5.58) | 9.56 (45.87) | 24.39 |
| RL | 31.76 (12.81) | 97.13 (2.85) | 41.25 (4.36) | 68.24 (12.81) | 8.21 |
| GA | 77.98 (33.41) | 93.73 (6.26) | 48.67 (3.06) | 22.02 (33.41) | 19.03 |
| IU | 95.85 (51.28) | 95.46 (4.52) | 55.67 (10.06) | 4.15 (51.28) | 29.29 |
| BE | 94.89 (50.32) | 94.94 (5.05) | 51.43 (5.82) | 5.11 (50.32) | 27.88 |
| BS | 94.33 (49.76) | 94.59 (5.39) | 50.81 (5.20) | 5.67 (49.76) | 27.53 |
| $l_1$-sparse | 95.83 (51.26) | 95.46 (4.53) | 55.17 (9.56) | 4.17 (51.25) | 29.15 |
| SFRon | 51.20 (6.63) | 99.79 (0.19) | 45.49 (0.12) | 48.80 (6.63) | 3.39 |
| SalUn | 39.88 (4.69) | 99.95 (0.03) | 43.47 (2.14) | 60.12 (4.69) | 2.89 |
| **FalW** | 46.94 (2.37) | 99.98 (0.00) | 46.11 (0.50) | 53.06 (2.37) | **1.31** |

| Method | Tiny-Imagenet, Vgg-16 , $\gamma = 1$ | | | | |
|---|---|---|---|---|---|
| | Random Forget (10%) | | | | |
| | FA | RA | TA | MIA | Avg. Gap |
| Retrain | 25.63 (0.00) | 99.98 (0.00) | 48.83 (0.00) | 74.36 (0.00) | 0.00 |
| FT | 86.97 (61.33) | 99.98 (0.00) | 53.95 (5.12) | 13.03 (61.33) | 31.95 |
| RL | 18.50 (7.13) | 99.83 (0.15) | 48.51 (0.32) | 81.50 (7.14) | 3.69 |
| GA | 77.98 (52.34) | 93.73 (6.26) | 48.67 (0.16) | 22.02 (52.34) | 27.77 |
| IU | 96.29 (70.66) | 95.46 (4.52) | 55.71 (6.88) | 3.71 (70.65) | 38.18 |
| BE | 74.92 (49.29) | 92.61 (7.37) | 45.89 (2.94) | 25.08 (49.28) | 27.22 |
| BS | 72.60 (46.97) | 92.42 (7.57) | 46.25 (2.58) | 27.40 (46.96) | 26.02 |
| $l_1$-sparse | 96.33 (70.70) | 95.47 (4.52) | 55.19 (6.36) | 3.67 (70.69) | 38.07 |
| SFRon | 22.01 (3.62) | 97.10 (2.88) | 44.23 (4.60) | 77.98 (3.62) | 3.68 |
| SalUn | 22.68 (2.95) | 99.18 (0.80) | 46.11 (2.72) | 71.02 (3.34) | 2.45 |
| **FalW** | 25.70 (0.07) | 99.98 (0.00) | 49.11 (0.28) | 73.30 (1.06) | **0.35** |

Table 6: Results on the **Cifar-10** dataset. (·) indicates performance gaps between approximate unlearning methods and the Retrain method across metrics.

| Method | CIFAR-10, Vgg-16 , $\gamma = 1/2$ | | | | |
|---|---|---|---|---|---|
| | Random Forget (40%) | | | | |
| | FA | RA | TA | MIA | Avg. Gap |
| Retrain | 65.94 (0.00) | 100.00 (0.00) | 78.62 (0.00) | 34.06 (0.00) | 0.00 |
| FT | 68.42 (2.48) | 96.51 (3.48) | 78.37 (0.25) | 31.58 (2.48) | 2.18 |
| RL | 72.37 (6.43) | 98.87 (1.13) | 81.15 (2.53) | 27.63 (6.43) | 4.13 |
| GA | 58.67 (7.27) | 84.28 (15.71) | 67.21 (11.41) | 41.33 (7.27) | 10.42 |
| IU | 98.84 (32.90) | 99.20 (0.80) | 90.47 (11.85) | 1.16 (32.90) | 19.61 |
| BE | 91.61 (25.67) | 98.26 (1.74) | 85.38 (6.76) | 8.39 (25.67) | 14.96 |
| BS | 82.61 (16.67) | 95.93 (4.06) | 80.83 (2.21) | 17.39 (16.67) | 9.90 |
| $l_1$-sparse | 82.58 (16.64) | 95.44 (4.55) | 82.95 (4.33) | 17.42 (16.64) | 10.54 |
| SFRon | 68.08 (2.14) | 97.10 (2.90) | 82.15 (3.53) | 30.91 (3.15) | 2.93 |
| SalUn | 63.84 (2.09) | 96.04 (3.96) | 77.03 (1.59) | 33.16 (0.91) | 2.14 |
| **FalW** | 65.64 (0.30) | 99.99 (0.00) | 78.01 (0.61) | 33.36 (0.70) | **0.40** |

Table 7: Plug-and-Play Experimental Results for FaLW. Comparison of baseline methods before and after FaLW integration under CIFAR-100, ResNet-18, $\gamma = 1/4$, 20% Random Forget setting.

| Method | FA | RA | TA | MIA | Avg.Gap |
|--------|------|------|------|------|---------|
| Retrain | 62.09 | 99.95 | 62.10 | 37.91 | 0.00 |
| GA | 95.22 | 96.66 | 68.83 | 4.78 | 19.07 |
| **GA + FaLW** | 89.678 | 90.625 | 63.48 | 10.32 | **16.47** |
| RL | 47.49 | 96.99 | 61.46 | 42.51 | 5.70 |
| **RL + FaLW** | 62.244 | 99.969 | 66.46 | 37.75 | **1.17** |
| SFRon | 65.30 | 99.98 | 66.42 | 33.69 | 2.95 |
| **SFRon + FaLW** | 62.856 | 99.653 | 62.77 | 37.144 | **0.62** |
| SalUn | 59.71 | 96.92 | 60.11 | 40.29 | 2.44 |
| **SalUn + FaLW** | 61.74 | 99.98 | 62.69 | 38.26 | **0.33** |

Table 8: Comparison between FaLW-normal and FaLW-beta variants on CIFAR-100 and Tiny-ImageNet.

| Dataset (Setting) | Method | FA | RA | TA | MIA | Avg. Gap $\downarrow$ |
|-------------------|--------|------|------|------|------|---------|
| **CIFAR-100** | Retrain | 62.09 | 99.95 | 62.10 | 37.91 | 0.00 |
| (ResNet-18, $\gamma = 1/4$) | **FaLW-normal** | 61.74 | 99.98 | 62.69 | 38.26 | **0.33** |
| (20% Forget) | FaLW-beta | 59.96 | 99.95 | 63.06 | 39.04 | 1.06 |
| **Tiny-ImageNet** | Retrain | 25.63 | 99.98 | 48.83 | 74.36 | 0.00 |
| (VGG-16, $\gamma = 1$) | **FaLW-normal** | 25.70 | 99.98 | 49.11 | 73.30 | **0.35** |
| (10% Forget) | FaLW-beta | 26.91 | 99.97 | 48.77 | 72.09 | 0.91 |

