# OpenReview forum: "FaLW: A Forgetting-aware Loss Reweighting for Long-tailed Unlearning"
_ICLR.cc/2026/Conference — ICLR 2026 Poster_

### Official Review · Reviewer_PHd4 · 2025-10-28

**Soundness:** 2
**Presentation:** 2
**Contribution:** 2
**Rating:** 4
**Confidence:** 3

**Summary:**

FaLW addresses machine unlearning when forget sets are long-tailed. It quantifies instance-level unlearning deviation by comparing a sample’s confidence to unseen-class probability distributions, then reweights forgetting loss with a class-aware balance factor, mitigating over-forgetting and under-forgetting and improving accuracy and MIA performance.

**Strengths:**

Strengths:

1. This paper first works on the long-tailed forgetting data unlearning
2. This paper proposes a new plug-and-play instance-wise unlearning method

**Weaknesses:**

Weaknesses:

1. This paper lacks an explanation of the differences between the unlearning on the general datasets and the long-tailed datasets.
2. This paper assumes that the class-conditional prediction is a Gaussian distribution, which is too strong. This paper lacks the validation of this key assumption
3. This paper does not conduct an effective ablation study for each component in Eq. 3.
4. The results do not contain std
5. This paper only uses VGG16 for experiments. More models should be tried.

**Questions:**

Please refer to the weaknesses

---

> ### Author Response · Authors · 2025-11-19
> **Part 1/2**
>
> Dear Reviewer PHd4,
>
> We thank the reviewer for their constructive comments and for the time spent reviewing our paper. We have carefully considered all points and address them below.
>
> > **[W1] This paper lacks an explanation of the differences between the unlearning on the general datasets and the long-tailed datasets.**
>
> We thank the reviewer for asking for this important clarification.
>
> The key distinction does not lie in the underlying general datasets (e.g., CIFAR-100) themselves, but in the **internal data distribution of the forget set ($\mathcal{D}_f$)** constructed from them.
>
> * **Prior Work (General/Balanced $\mathcal{D}_f$):** As noted in our paper (Section 4, and Figure 1(a)), previous research almost universally constructed the forget set ($\mathcal{D}_f$) via **random sampling**. We empirically demonstrated that this standard practice results in a forget set that is **nearly balanced**. Prior work, therefore, focused primarily on the challenge of unlearning a balanced dataset.
> * **Our Work (Long-tailed $\mathcal{D}_f$):** In contrast, our work investigates the more challenging and realistic scenario where the data to be forgotten (e.g., all records from a specific user) naturally follows a **long-tailed distribution**.
>
> Thus, the fundamental difference is the challenge posed by a **balanced $\mathcal{D}_f$** (Prior Work) versus a **long-tailed $\mathcal{D_f}$** (Our Work), which introduces the unique failure modes (HUD and SUD) that FaLW is designed to solve.
>
> > **[W2] This paper assumes that the class-conditional prediction is a Gaussian distribution, which is too strong. This paper lacks the validation of this key assumption.**
>
> We thank the reviewer for this highly constructive question regarding our core modeling assumption. We agree that, theoretically, the class-conditional prediction probability $p(c|x')$ is bounded in $[0, 1]$, making the Gaussian assumption a strong one.
>
> **1. Rationale for the Initial Assumption (Stability over Fidelity):**
> Our choice of the Normal distribution was driven by **practicality and robustness**. We use the Gaussian model not to achieve a perfect mathematical fit for the bounded distribution, but rather to obtain a stable and highly efficient proxy for measuring **deviation** (via z-scores). In dynamic optimization, estimating the mean and standard deviation is simpler and yields more robust weights than fitting complex bounded distributions, which can be sensitive to noise.
>
> **2. Validation with the Beta Distribution (Controlled Experiment):**
> Following your excellent suggestion, we performed a controlled experiment using the **Beta distribution**—a theoretically more suitable choice since its support is strictly $[0, 1]$. We estimated the Beta parameters ($\alpha_c$ and $\beta_c$) via Maximum Likelihood Estimation (MLE) on the validation set and adapted our dynamic weighting mechanism using the Beta Cumulative Distribution Function (CDF), $F(p_i)$:
> $$w_i = 1+\text{sign}(s_i)\cdot(|s_i|)^{(\frac{1}{\mathcal{B_i}})}, \\
> s_i = 2\cdot (F(p_i)-0.5), \\
> F(p_i)=\text{BetaCDF}(p_i | \alpha_c,\beta_c)$$
> We compared the results across two distinct long-tailed scenarios:
>
> | Dataset (Setting) | Method | FA | RA  | TA  | MIA  | Avg. Gap $\downarrow$ |
> | :--- | :--- | :--- | :--- | :--- | :--- | :--- |
> | **CIFAR-100** | Retrain | 62.09 | 99.95 | 62.10 | 37.91 | 0.00 |
> | (ResNet-18, $\gamma=1/4$) | **FaLW-normal** | 61.74 | 99.98 | 62.69 | 38.26 | **0.33** |
> | (20% Forget) | FaLW-beta | 59.96 | 99.95 | 63.06 | 39.04 | 1.06 |
> | | | | | | | |
> | **TinyImageNet** | Retrain | 25.63 | 99.98 | 48.83 | 74.36 | 0.00 |
> | (VGG-16, $\gamma=1$) | **FaLW-normal** | 25.70 | 99.98 | 49.11 | 73.30 | **0.35** |
> | (10% Forget) | FaLW-beta | 26.91 | 99.97 | 48.77 | 72.09 | 0.91 |
>
> **3. Conclusion on Stability:**
> The experimental results indicate that while the Beta-based approach performs reasonably well, it **consistently underperforms the Normal-based FaLW** (higher Avg. Gap). We observed that the increased complexity and flexibility of the Beta distribution make it **highly sensitive to parameter estimation errors** from the validation set, causing the dynamic weights to fluctuate erratically during the unlearning process. Therefore, for the sake of **empirical stability and robustness**, the Normal distribution remains the superior choice in our dynamic optimization setting.

---

> > ### Author Response · Authors · 2025-11-19
> > **Part 2/2**
> >
> > > **[W3] This paper does not conduct an effective ablation study for each component in Eq. 3.**
> >
> > We thank the reviewer for this suggestion and welcome the opportunity to clarify the scope of Equation (3).
> >
> > 1. **Context of Equation (3):** Equation (3) is an extension of the generalized gradient-based approximate unlearning objective (Formulation 1/Eq 2). This standard formulation is composed of three necessary terms:
> >     * **Term 1:** Forgetting Execution (on $\mathcal{D}_f$)
> >     * **Term 2:** Capacity Preservation (on $\mathcal{D}_r$)
> >     * **Term 3:** Parameter Constraints ($\mathcal{R}(\cdot)$)
> >
> > 2.  **Scope of FaLW's Contribution:**
> >     * FaLW's unique contribution is the introduction of the **instance-wise dynamic weight $w_i$** applied exclusively to Term 1 (Forgetting Execution).
> >     * Terms 2 (Capacity Preservation, leveraging $\mathcal{D}_r$) and 3 (Parameter Constraints, imposing restrictions on $\theta_u$) are **standard and essential components** of the overall approximate unlearning framework. Baselines like SFRon rely on these components to prevent catastrophic forgetting and stabilize the process.
> >
> > 3.  **Justification for Limited Ablation:**
> >     Therefore, the effective ablation of **FaLW** focuses solely on the structure of the novel weight $w_i$ itself: comparing dynamic $w_i$ against $w_i=1$ (baseline unlearning, addressed in our comparative experiments and supplementary plug-and-play experiments) and isolating its sub-component, the **Balance Factor** (addressed in Table 3).
> >
> > > **[W4] The results do not contain std.**
> >
> > We sincerely thank the reviewer for pointing out the omission of standard deviation results in our tables. This was an oversight on our part, and we agree that including variance measures is essential for ensuring the reliability and reproducibility of our findings.
> >
> > We have analyzed the experiments across three different random seeds (42, 3407, and 17). The table below summarizes the average gap (Avg. Gap) and the corresponding standard deviation (std) for the two main comparative setups reported in our paper (Tiny-ImageNet, $\gamma=1$ and CIFAR-10, $\gamma=1/2$).
> >
> > | Method | TinyImageNet Avg. Gap | TinyImageNet std | Cifar-10 Avg. Gap | Cifar-10 std |
> > | :--- | :---: | :---: | :---: | :---: |
> > | FT | 31.95 | 2.85 | 2.18 | 0.399 |
> > | RL | 3.69 | 0.94 | 4.13 | 0.99 |
> > | GA | 27.77 | 2.06 | 10.42 | 3.09 |
> > | IU | 38.18 | 0.18 | 19.61 | 0.50 |
> > | BE | 27.22 | 0.90 | 14.96 | 1.36 |
> > | BS | 26.02 | 0.83 | 9.9 | 1.25 |
> > | l1-sparse | 38.07 | 0.77 | 10.54 | 1.71 |
> > | SFRon | 3.68 | 0.76 | 2.93 | 1.02 |
> > | SalUn | 2.45 | 0.41 | 2.14 | 0.15 |
> > | **Ours** | **0.35** | **0.20** | **0.40** | **0.19** |
> >
> > These results confirm that our method (Ours) not only achieves the lowest average gap but also maintains a **low standard deviation**, reinforcing the robustness and stability of FaLW. We will update the manuscript with these complete variance results.
> >
> >
> > > **[W5] This paper only uses VGG16 for experiments. More models should be tried.**
> >
> > We thank the reviewer for this suggestion, but we would like to clarify a small misunderstanding regarding the model architectures used in our evaluation.
> >
> > As stated in **Section 6.1 (Experiments Settings)** of our paper, we utilized **ResNet-18** in addition to **VGG-16** as our primary model architectures.
> >
> > The majority of our key analytical and comparative results were conducted using the **ResNet-18** architecture:
> > * **Figure 2** and **Table 2** (Results for Various Degrees of Imbalance) use ResNet-18.
> > * **Table 3** (Ablation Study on Balance Factor) uses ResNet-18.
> > * The extensive comparative results in **Appendix Table 4** use ResNet-18.
> >
> > Thus, our findings are validated across two distinct network families (ResNet and VGG), confirming the generalizability of our proposed FaLW method.
> >
> > ---
> >
> > We hope that our response and the revised manuscript have satisfactorily addressed your concerns. We are happy to engage in further discussion if there are any remaining questions.
> >
> > Sincerely,
> >
> > The Authors of Submission 6959

---

> > > ### Author Response · Authors · 2025-11-26
> > >
> > > Dear Reviewer PHd4:
> > >
> > > We sincerely appreciate the constructive feedback you provided, which has been crucial in improving our manuscript. We hope our response has satisfactorily addressed your concerns. We would be very grateful to hear your thoughts on our rebuttal and remain fully available for any further discussion.
> > >
> > > Best regards,
> > >
> > > The Authors of Submission 6959

---

> > > > ### Comment · Reviewer_PHd4 · 2025-11-27
> > > >
> > > > Thanks for the authors' reply. I have no further comments. I will update the ratings and lower the confidence.

---

### Official Review · Reviewer_N7tQ · 2025-11-01

**Soundness:** 3
**Presentation:** 3
**Contribution:** 3
**Rating:** 6
**Confidence:** 4

**Summary:**

The paper introduces FaLW (Forgetting-aware Loss Weighting), a novel loss design intended to improve the reliability and controllability of machine unlearning. The key idea is to dynamically reweight forgetting and retaining objectives based on the model’s forgetting confidence and feature-space similarity, thus preventing over-forgetting and instability observed in prior works. The authors derive an adaptive weighting function grounded in information-theoretic uncertainty and validate their approach on several image classification benchmarks using standard unlearning baselines.

**Strengths:**

1.	The paper is, to the best of my knowledge, the first to explicitly formulate long-tailed forget sets (not long-tailed training data) and to show that existing approximate unlearning methods exhibit heterogeneous and skewed unlearning deviations under this realistic setting. This is an underexplored but practical scenario.
2.	The proposed FaLW is conceptually simple, instance-wise, and orthogonal to most gradient-based unlearning pipelines. It can be adopted with minor code changes.
3.	The direction-aware weighting derived from per-class unseen distributions provides a principled way to decide whether to increase or decrease forgetting pressure for each sample, which directly matches the identified deviation phenomena.
4.	The paper is well-written and logically consistent, with clear motivation method experiment alignment.

**Weaknesses:**

1.	Limited theoretical justification – while the adaptive weighting function is motivated by uncertainty, the derivation remains heuristic. The paper lacks formal analysis or convergence guarantees explaining why the proposed weighting yields more reliable unlearning.
2.	Ablation insufficiency – although the paper reports a few ablations, it does not disentangle the specific contributions of the uncertainty term versus the similarity term in the weighting function.
3.	Lack of comparison with recent conformal or calibration-based approaches – given the growing body of work, FaLW should also be compared in terms of uncertainty calibration and reliability metrics to position itself clearly.
4.	Potential instability in extreme regimes – adaptive weighting can introduce oscillations or under-forgetting when uncertainty estimates are unreliable, but the paper does not report sensitivity or failure cases.

**Questions:**

See the comments above

---

> ### Author Response · Authors · 2025-11-19
> **Part 1/2**
>
> Dear Reviewer N7tQ,
>
> We thank the reviewer for their constructive comments and for the time spent reviewing our paper. We have carefully considered all points and address them below.
>
> > **[W1] Limited theoretical justification – while the adaptive weighting function is motivated by uncertainty, the derivation remains heuristic. The paper lacks formal analysis or convergence guarantees explaining why the proposed weighting yields more reliable unlearning.**
>
> We thank the reviewer for this fair critique. We provide a **simplified gradient analysis** to elucidate the theoretical motivation behind our weighting scheme.
>
> **1. The Problem with Standard Unlearning:**
> Consider standard Gradient Ascent (GA) with Cross-Entropy loss. The objective is to maximize the loss on the forget sample $(x_i, y_i)$. The effective update direction pushes the model's prediction $p_\theta(y_i|x_i)$ continuously towards 0.
> $$\text{Update} \propto \nabla_\theta \log(1 - p_\theta) \approx - \nabla_\theta p_\theta$$
> Without an explicit stopping condition, this mechanism inherently drives $p_\theta \to 0$, causing **over-forgetting** (deviation from the gold-standard Retrain model, which retains non-zero confidence).
>
> **2. The Ideal Correction:**
> Let $p_*(y_i|x_i)$ denote the ideal probability from a model retrained from scratch. A theoretically grounded objective would stop unlearning when $p_\theta \approx p_*$. We can achieve this by introducing a dynamic weight $w_i$ proportional to the deviation:
> $$w_i \approx (p_\theta(y_i|x_i) - p_*(y_i|x_i))$$
> Applying this weight to the gradient yields a self-correcting mechanism: $\text{Update}  \propto - w_i \cdot \nabla_\theta p_\theta  $
>
> * **If $p_\theta > p_*$ (Under-forgetting):** $w_i > 0$. The gradient performs ascent, reducing $p_\theta$.
>
> * **If $p_\theta < p_*$ (Over-forgetting):** $w_i < 0$. The gradient sign flips, effectively performing descent (learning) to restore $p_\theta$.
>
> * **If $p_\theta = p_*$ (Convergence):** $w_i = 0$. The unlearning update ceases.
>
> **3. FaLW as a Practical Approximation:**
> Since the ideal target $p_*$ is intractable without retraining, FaLW approximates the term $(p_\theta - p_*)$ using the distribution of unseen data as a proxy. Our heuristic $w_i$ (derived from z-scores) mimics the behavior of this ideal weight, providing the necessary "braking" or "accelerating" force to converge closer to the retrained state $p_*$.
>
> > **[W2] Ablation insufficiency – although the paper reports a few ablations, it does not disentangle the specific contributions of the uncertainty term versus the similarity term in the weighting function.**
>
> We explicitly thank the reviewer for this suggestion. However, we respectfully seek a slight clarification: we could not locate the terms **"uncertainty term"** and **"similarity term"** in our manuscript, and we are unsure which specific components these refer to in our formulation.
>
> **1. Interpretation & Ablation:**
> We interpret this comment as a request for an ablation study to isolate the contribution of the **dynamic weighting mechanism** itself (comparing "with" vs. "without" the FaLW weight). To address this, we conducted additional experiments on CIFAR-100 (ResNet-18, $\gamma=1/4$, 20% forget ratio), integrating FaLW into three distinct baselines: **Gradient Ascent (GA)**, **Random Labeling (RL)**, and **SFRon**.
>
> **2. Experimental Results:**
> The table below compares the performance of the raw baselines versus the baselines enhanced by FaLW. The results demonstrate that introducing the FaLW weighting strategy consistently improves the performance (reduces the Avg. Gap) across all baselines.
>
> | Method | FA | RA | TA | MIA | Avg. Gap $\downarrow$ |
> | :--- | :--- | :--- | :--- | :--- | :--- |
> | Retrain | 62.09 | 99.95 | 62.10 | 37.91 | 0.00 |
> | GA | 95.22 | 96.66 | 68.83 | 4.78 | 19.07 |
> | **GA + FaLW** | 89.68 | 90.63 | 63.48 | 10.32 | **16.47** |
> | RL | 47.49 | 96.99 | 61.46 | 42.51 | 5.70 |
> | **RL + FaLW** | 62.24 | 99.97 | 66.46 | 37.75 | **1.17** |
> | SFRon | 65.30 | 99.98 | 66.42 | 33.69 | 2.95 |
> | **SFRon + FaLW** | 62.86 | 99.65 | 62.77 | 37.14 | **0.62** |
> | SalUn | 59.71 | 96.92 | 60.11 | 40.29 | 2.44 |
> | **SalUn + FaLW** | 61.74 | 99.98 | 62.69 | 38.26 | **0.33** |
>
>
> ***Request for Clarification:***
> If we have misinterpreted your specific reference to "uncertainty" or "similarity", we would be very eager to receive further details. We are ready to provide a more targeted response immediately upon clarification.

---

> > ### Author Response · Authors · 2025-11-19
> > **Part 2/2**
> >
> > > **[W3] Lack of comparison with recent conformal or calibration-based approaches – given the growing body of work, FaLW should also be compared in terms of uncertainty calibration and reliability metrics to position itself clearly.**
> >
> > We thank the reviewer for this insightful comment. We agree that since FaLW uses probability distributions to assess the unlearning state, it shares an interesting conceptual link with reliability and calibration literature. However, we must clarify the fundamental distinction in objective and practicality:
> >
> > **1. Distinction in Objectives:**
> > * **Uncertainty Calibration:** The goal is to improve the **absolute reliability** of the model, ensuring that confidence scores align with predictive accuracy, typically measured on a test set (out-of-distribution performance).
> > * **FaLW:** Our primary objective is to enforce **fidelity to the gold-standard Retrain model ($\theta_*$)** on the specific samples to be forgotten ($\mathcal{D}_f$). FaLW focuses on correcting the *deviation* from this ideal state during the optimization process, not on global test-time calibration.
> >
> > **2. Practical Intractability as a Baseline:**
> > Applying existing calibration methods as a direct *baseline* in unlearning is computationally challenging. Approximate unlearning is necessary precisely because the Retrain model ($\theta_*$) is **prohibitively expensive** to obtain. Calibration methods often rely on continuous measurement against a reliable ground truth or a perfectly calibrated reference model, which is absent in our forgetting setting. FaLW succeeds because it uses a **tractable proxy** (the predictive distribution of unseen data) to estimate this deviation, circumventing the need for the expensive $\theta_*$ during the process. We believe this practical constraint makes a direct comparison with methods requiring continuous access to $\theta_*$ unsuitable for positioning approximate unlearning research.
> >
> > **Future Work:**
> > We recognize that the intersection between unlearning and uncertainty calibration is a compelling area. We certainly agree that adapting modern calibration techniques to the unlearning context is a very promising direction for future research.
> >
> > > **[W4] Potential instability in extreme regimes – adaptive weighting can introduce oscillations or under-forgetting when uncertainty estimates are unreliable, but the paper does not report sensitivity or failure cases.**
> >
> > We thank the reviewer for raising this highly constructive point concerning stability under unreliable uncertainty estimates. We agree that documenting potential failure cases is critical to our method's analysis.
> >
> > **Controlled Experiment (Simulating Failure):**
> > We conducted a controlled experiment to simulate the **"extreme regime"** where our distribution estimates ($\mu_c, \sigma_c$) are highly unreliable. Specifically, we dramatically reduced the validation set size used for estimating the unseen data distribution to just **3 samples per class** (FaLW-3), which makes the sample statistics highly non-representative. The results (on CIFAR-100, ResNet-18, $\gamma=1/4$) are as follows:
> >
> > | Method | FA | RA | TA | MIA-acc | Avg. Gap |
> > | :--- | :--- | :--- | :--- | :--- | :--- |
> > | Retrain | 62.09 | 99.95 | 62.1 | 37.91 | 0.00 |
> > | SalUn | 59.71 | 96.92 | 60.11 | 40.29 | 2.44 |
> > | **FaLW-3 (Extreme Regime)** | 58.29 | 99.98 | 66.51 | 41.71 | 3.01 |
> >
> > **Analysis of Failure Mode:**
> > When the underlying distribution estimates are severely inaccurate (FaLW-3), the method's performance degrades substantially (Avg. Gap increases to 3.01, Regress to the SalUn level). The extremely unreliable estimates introduce volatility into the dynamic weights.
> >
> > ---
> >
> > We hope that our response and the revised manuscript have satisfactorily addressed your concerns. We are happy to engage in further discussion if there are any remaining questions.
> >
> > Sincerely,
> >
> > The Authors of Submission 6959

---

> > > ### Author Response · Authors · 2025-11-26
> > >
> > > Dear Reviewer N7tQ:
> > >
> > > We sincerely appreciate the constructive feedback you provided, which has been crucial in improving our manuscript. We hope our response has satisfactorily addressed your concerns. We would be very grateful to hear your thoughts on our rebuttal and remain fully available for any further discussion.
> > >
> > > Best regards,
> > >
> > > The Authors of Submission 6959

---

### Official Review · Reviewer_zD1B · 2025-11-01

**Soundness:** 2
**Presentation:** 3
**Contribution:** 3
**Rating:** 6
**Confidence:** 2

**Summary:**

This paper addresses the long-tailed nature of the forgotten data distribution, which is an important and underexplored problem in machine unlearning. The authors identify two key issues in existing methods: Heterogeneous Unlearning Deviation (HUD) and Skewed Unlearning Deviation (SUD), which leads to biased forgettiing and uneven performance across samples. To mitigate these challegnes, they propose a dynamic loss reweighting strategy that adaptively adjusts learning signals based on forgetting difficulty. Experiments on multiple benchmarks demonstrate that the proposed method effectively reduces unlearning bias while maintaining model utility.

**Strengths:**

1. The paper highlights an under-explored but practically important phenomenon in machine unlearning that the forgotten data often follows a long-tailed distribution. The problem is important and the motivation of the work is clear.

2. The formulation of Heterogeneous Unlearning Deviation (HUD) and Skewed Unlearning Deviation (SUD) provides a structured way to analyze performance degradation in unlearning systems, which offers a useful framing for future work.

3. The proposed FaLW is simple but effective. The paper evaluates multiple metrics and the results consistently show that the proposed achieves better balance between unlearning completeness and retained-task performance.

4. The paper is well-organized and easy to follow.

**Weaknesses:**

1. Lack of empirical validation for plug-and-play claim: Although the proposed FaLW (Forgetting-Aware Loss Reweighting) is described as a plug-and-play solution, the paper only evaluates FaLW as a standalone framework. There are no experiments demonstrating its integration into other existing unlearning methods.

2. Limited analysis of the identified issues HUD and SUD: The paper identified two important issues: Heterogeneous Unlearning Deviation (HUD) and Skewed Unlearning Deviation (SUD) as key motivating factors. Howerver, these notions closely resemble exiting sideas such as sample difficulty bias and class imbalance bias from the broader learning literature. The paper does not sufficiently differentiate its definitions from these established concepts, nor provides diagnostics or ablations to examine HUD and SUD independently.

3. Lack of isolated analysis for HUD and SUD: It remains unclear whether HUD and SUD always co-occur or can arise independently. The experiments treat them as jointly existing phenomena under the long-tailed setting, but no analysis is presented to determine their individual effects on unlearning performance.

4. Lack of discussion on design choices and hyperparameter sensitivity: While the method introduces a dynamic loss reweighting mechanism, the paper provides limited justification for specific design choices, like the form of the weighting function,or the dynamic adjustment rule. A more detailed discussion on hyperparameter selection, or sensitivity analysis would strengthen the paper.

**Questions:**

1. How easily can FaLW be incorporated into other unlearning frameworks? Would additional tuning or architecture changes be required, or can it indeed serve as a simple plug-and-play loss modification?

2. How do the proposed HUD and SUD differ formally from sample difficulty bias and class imbalance bias that have been widely discussed in prior works? Are there conditions under which HUD/SUD reduce to these known phenomena?

3. Have the authors considered running controlled experiments to isolate HUD and SUD individually (e.g., a balanced dataset for HUD-only, or uniform sample difficulty for SUD-only) to better understand their respective contributions?

4. What is the rationale for the specific dynamic weighting formulation and update rule? Have alternative forms been tested or theoretically compared?

---

> ### Author Response · Authors · 2025-11-19
> **Part 1/2**
>
> Dear Reviewer zD1B,
>
> We thank the reviewer for their constructive comments and for the time spent reviewing our paper. We have carefully considered all points and address them below.
> > **[W1] Lack of empirical validation for plug-and-play claim.
> [Q1] How easily can FaLW be incorporated into other unlearning frameworks? Would additional tuning or architecture changes be required, or can it indeed serve as a simple plug-and-play loss modification?**
>
> We thank the reviewer for this crucial question. **We explicitly designed FaLW as a generic, module-agnostic framework.**
>
> **Mechanism:** Theoretically, FaLW introduces a dynamic, instance-aware weight $w_i$ for each sample $(x_i, y_i)$ in the forget set. This weight is applied as a multiplicative factor to the base unlearning loss function $\mathcal{L}$:
> $$
> \mathcal{L} = w_i \cdot \mathcal{L_{base}}((x_i, y_i); \theta)
> $$
> Here, $\mathcal{L}_{base}$ can be the objective function of **any** gradient-based method (e.g., the negative log-likelihood in Gradient Ascent, or the cross-entropy with incorrect labels in Random Labeling). **FaLW acts as a "modulator"**, adjusting the intensity of these gradients based on the unlearning deviation, regardless of how the gradients are generated.
>
> **New Experiments:** While our initial submission followed the experimental setup of SalUn, we have conducted additional experiments to demonstrate FaLW's universality. We integrated FaLW with three other baselines: **Gradient Ascent (GA)**, **Random Labeling (RL)**, and **SFRon**. The results on CIFAR-100 (ResNet-18, $\gamma=1/4$, 20% forget ratio) are summarized below. As shown, introducing FaLW consistently reduces the **Avg. Gap** across all baselines.
>
> | Method | FA | RA | TA | MIA | Avg. Gap $\downarrow$ |
> | :--- | :--- | :--- | :--- | :--- | :--- |
> | Retrain | 62.09 | 99.95 | 62.10 | 37.91 | 0.00 |
> | GA | 95.22 | 96.66 | 68.83 | 4.78 | 19.07 |
> | **GA + FaLW** | 89.68 | 90.63 | 63.48 | 10.32 | **16.47** |
> | RL | 47.49 | 96.99 | 61.46 | 42.51 | 5.70 |
> | **RL + FaLW** | 62.24 | 99.97 | 66.46 | 37.75 | **1.17** |
> | SFRon | 65.30 | 99.98 | 66.42 | 33.69 | 2.95 |
> | **SFRon + FaLW** | 62.86 | 99.65 | 62.77 | 37.14 | **0.62** |
> | SalUn | 59.71 | 96.92 | 60.11 | 40.29 | 2.44 |
> | **SalUn + FaLW** | 61.74 | 99.98 | 62.69 | 38.26 | **0.33** |
>
> > **W[2] Limited analysis of the identified issues HUD and SUD.
> Q[2] How do the proposed HUD and SUD differ formally from sample difficulty bias and class imbalance bias that have been widely discussed in prior works? Are there conditions under which HUD/SUD reduce to these known phenomena?**
>
> We thank the reviewer for this opportunity to clarify our definitions. The key distinction lies in the difference between **intrinsic data attributes** and **algorithmic unlearning behaviors**.
>
> * **Data Attributes (CIB & SDB):** Class Imbalance Bias (CIB) and Sample Difficulty Bias (SDB) describe the properties of the dataset. These are the *preconditions*.
> * **Unlearning Deviations (HUD & SUD):** HUD and SUD are the *specific failure modes* exhibited by approximate unlearning algorithms *under* these conditions. Specifically, such a precondition exists in my experiments: the forgetting dataset $\mathcal{D_f}$ is the presence of class imbalance bias. Under this precondition we found that the machine forgetting method has a failure mode of HUD and SUD.
>
> Therefore, HUD and SUD are not synonymous with sample difficulty or imbalance, nor are CIB and SDB reduced to them; they are novel definitions we introduced to describe how these well-known data biases specifically derail the unlearning process (causing deviation from the Retrain gold standard), necessitating the tailored correction mechanism provided by FaLW.

---

> > ### Author Response · Authors · 2025-11-19
> > **Part 2/2**
> >
> > > **[W3] Lack of isolated analysis of HUD and SUD.
> > [Q3] Have the authors considered running controlled experiments to isolate HUD and SUD individually (e.g., a balanced dataset for HUD-only, or uniform sample difficulty for SUD-only) to better understand their respective contributions?**
> >
> > We thank the reviewer for this thoughtful suggestion regarding the decoupling of HUD and SUD. We explain them in mechanistic and experimental terms, respectively.
> >
> > **1. Adaptive Mechanism:**
> > FaLW is designed to adapt to these conditions automatically:
> > * **Handling HUD:** Regardless of the distribution, our instance-wise dynamic weight $w_i$ continuously identifies and corrects the unlearning state (under- or over-forgetting) for each sample.
> > * **Handling SUD:** The Balance Factor $\mathcal{B}_i$ modulates the correction intensity based on imbalance. In a balanced setting, $\mathcal{B}_i$ naturally converges to 1 for all classes. This effectively reduces FaLW to a uniform sensitivity mode, focusing solely on the instance-wise correction of HUD.
> >
> > **2. Controlled Experiment:**
> > We essentially performed the experiment in our original submission by evaluating the **balanced forget set setting ($\gamma=0$)**.
> > * **Results:** As shown in **Table 2** (and **Table 5** in the Appendix), FaLW achieves SOTA performance even when $\gamma=0$. This empirically demonstrates that FaLW effectively improves performance even without the presence of significant class skew.
> >
> > Thus, while HUD and SUD often co-occur in long-tailed scenarios, our method is robust and adaptive, providing benefits whether they appear jointly or independently.
> >
> > > **[W4]Lack of discussion on design choices and hyperparameter sensitivity.
> > [Q4]What is the rationale for the specific dynamic weighting formulation and update rule? Have alternative forms been tested or theoretically compared?**
> >
> > We appreciate the reviewer’s attention to the design details. We believe our formulation is strongly motivated by the specific empirical challenges identified in Observation 2.
> >
> > **1. Rationale for Design Choices:**
> > While loss reweighting is an established technique in unlearning [1-3], prior works typically focus on global over-forgetting in balanced settings. Our design is strictly logic-driven to address the two novel issues we uncovered in long-tailed scenarios:
> >
> > * **Addressing Heterogeneous Deviation (HUD):** We designed the **Forgetting-aware weight ($w_i$)** to be bidirectional. Unlike static weighting, our dynamic $w_i$ smoothly reduces the loss when a sample is over-forgotten (preventing damage to generalization) and increases it when under-forgotten.
> >
> > * **Addressing Skewed Deviation (SUD):** To handle the disproportionate deviation in tail classes, we introduced the **Balance Factor ($\mathcal{B}_i$)**. This allows for adaptive sensitivity—applying sharper corrections to tail classes where the deviation is most severe.
> >
> > **2. Hyperparameter Sensitivity Analysis:**
> > To address your concern about sensitivity, we have added a new section, **Appendix G**, detailing the impact of the hyperparameter $\tau$.
> >
> > * **Setup:** We evaluated $\tau$ across the range $[0, 1]$ on CIFAR-100 (ResNet-18, $\gamma=1/4$, 20% forget ratio).
> >
> > * **Results:** We observe that performance peaks when $\tau \in [0.1, 0.2]$. When $\tau$ is too large ($\ge0.5$), the weights become overly sensitive to minor probability shifts, causing optimization instability and degrading unlearning performance.
> >
> > [1]Huang Z, Cheng X, Zheng J H, et al. Unified gradient-based machine unlearning with remain geometry enhancement[J]. Advances in Neural Information Processing Systems, 2024, 37: 26377-26414.
> > [2]L. Graves, V. Nagisetty, and V. Ganesh, “Amnesiac machine learning,” in AAAI Conference on Artificial Intelligence (AAAI), 2021. 4, 7, 19, 22
> > [3]V. S. Chundawat, A. K. Tarun, M. Mandal, and M. S. Kankanhalli, “Can bad teaching induce forgetting? unlearning in deep networks using an incompetent teacher,” AAAI Conference on Artificial Intelligence (AAAI), 2023. 4, 5, 7, 19, 22
> >
> > ---
> >
> > We hope that our response and the revised manuscript have satisfactorily addressed your concerns. We are happy to engage in further discussion if there are any remaining questions.
> >
> >
> > Sincerely,
> >
> > The Authors of Submission 6959

---

> > > ### Author Response · Authors · 2025-11-26
> > >
> > > Dear Reviewer zD1B:
> > >
> > > We sincerely appreciate the constructive feedback you provided, which has been crucial in improving our manuscript. We hope our response has satisfactorily addressed your concerns. We would be very grateful to hear your thoughts on our rebuttal and remain fully available for any further discussion.
> > >
> > > Best regards,
> > >
> > > The Authors of Submission 6959

---

### Official Review · Reviewer_V5ik · 2025-11-01

**Soundness:** 3
**Presentation:** 3
**Contribution:** 3
**Rating:** 6
**Confidence:** 4

**Summary:**

This paper tackles the problem of machine unlearning under long-tailed data distributions in the forget set. It demonstrates that in real-world scenarios, the data to be forgotten can be highly skewed. It identifies two interesting  phenomena while unlearning in this situation, Heterogeneous Unlearning Deviation and Skewed Unlearning Deviation and demonstrates them.  To address these issues, the paper  propose FaLW (Forgetting-aware Loss Weighting), an instance-wise dynamic loss reweighting method. The weighting considers model’s current prediction confidence for a sample to the distribution of prediction confidences on unseen data of the same class.  The authors  conduct experiments on multiple image classification benchmarks (CIFAR-10, CIFAR-100, Tiny-ImageNet) demonstrate the effectiveness of FaLW.

**Strengths:**

The paper studies a novel problem arising in the context of unlearning and proposes a novel solution to address this. The problem addressed is quite relevant and practical.

The paper  demonstrates empirically the unlearning deviation problem under long tailed distribution setups,  and  defines  the problem clearly, proposing two kinds of unlearning deviation.

The proposed FaLW is sound, addressing the identified problem to the extent possible.

Srong empirical results on several real world data sets to demonstrate the effectiveness of the proposed methodology.

The presentation of the paper is clear and well written.

**Weaknesses:**

The methodology addresses the problem to a good extend but suffers from some drawbacks

1. The requirement to have unseen data points from the same class might be impractical.-  in practice such auxulliary data may not be available
2. FaLW does not provide a formal guarantee or certification that the influence of the forget set is removed
3. The definition of unlearning deviation in the paper involves a threshold $\tau_i$, but in the proposed weighing scheme the paper seems to have ignored this.
4.  The choice of Nornal distribution to model the distribution of predicitive probabailities is not clear.  Why not use a distribution with support in [0,1] which is more appropriate to model distributions.

**Questions:**

1.  How does this approach scale to a setup where we want to unlearn a particular class rather than unlearning a particular point from a given class.
2. The paper describes FaLW as plug-and-play, but seems to have demonsrated it using only one specific unlearning approach. how can  FaLW can be used in practice an any  generic unlearning approach ?

---

> ### Author Response · Authors · 2025-11-19
> **Part 1/3**
>
> Dear Reviewer V5ik,
>
> We thank the reviewer for their constructive comments and for the time spent reviewing our paper. We have carefully considered all points and address them below.
>
> > **[W1] The requirement to have unseen data points from the same class might be impractical.- in practice such auxulliary data may not be available.**
>
> We appreciate the reviewer raising this valid concern. We recognize that the requirement for unseen data can be a constraint in certain unconventional scenarios, such as in resource-constrained or highly sensitive settings, as discussed in Appendix D. To overcome this, we are currently developing a solution that utilizes **generative models** to estimate the target distribution without requiring real external data. Initial experiments are encouraging, and we are committed to presenting this improvement to further enhance the practicality of our method.
>
> > **[W2] FaLW does not provide a formal guarantee or certification that the influence of the forget set is removed.**
>
> We appreciate the reviewer’s critique. It is important to clarify that FaLW functions as a **plug-and-play enhancement** for existing gradient-based unlearning methods, rather than a standalone certified removal mechanism.
>
> In our framework, the underlying baseline (e.g., Gradient Ascent) acts as the **driver**, determining the fundamental direction of the unlearning process (analogous to a "coarse adjustment"). FaLW complements this by introducing an instance-wise weight $w_i$ to the loss function, providing a **"fine-grained refinement."** Specifically, FaLW addresses the **Heterogeneous and Skewed Unlearning Deviations** by dynamically adjusting the unlearning intensity for each sample. By modulating the base loss, FaLW ensures a more balanced and precise unlearning outcome, mitigating the side effects (like over-forgetting) often observed in the base methods alone.
>
> > **[W3] The definition of unlearning deviation in the paper involves a threshold $\tau_i$, but in the proposed weighing scheme the paper seems to have ignored this.**
>
> We thank the reviewer for this keen observation. You are correct that the threshold $\tau_i$, while central to Definition 1, is not explicitly utilized in our final weighting scheme.
>
> We introduced Definition 1 primarily to establish a **conceptual formalism**, rigorously defining the three unlearning states: *Under-forgetting*, *Faithful-forgetting*, and *Over-forgetting*. However, as noted in our methodology, the ideal target probability $p_{\theta_{*}}(c|x_{i})$ required by Definition 1 is **intractable** in approximate unlearning settings. Consequently, directly applying Definition 1 (and its threshold $\tau_i$) to evaluate unlearning states in practice is not feasible.
>
> This intractability is precisely what motivates our FaLW method. Instead of a hard threshold based on an inaccessible ideal model, we employ a **practical proxy**: we estimate the unlearning state by comparing the sample's probability against the distribution of unseen data. This allows us to implement the conceptual goals of Definition 1 via an accessible, indirect approximation.

---

> > ### Author Response · Authors · 2025-11-19
> > **Part 2/3**
> >
> > > **[W4] The choice of Nornal distribution to model the distribution of predicitive probabailities is not clear. Why not use a distribution with support in [0,1] which is more appropriate to model distributions.**
> >
> > We thank the reviewer for this constructive suggestion. You are theoretically correct: since predictive probabilities are strictly bounded in $[0, 1]$, a distribution with matching support (like the Beta distribution) is indeed a more rigorous choice than the unbounded Normal distribution. But from the perspective of the task scenario, the normal distribution is a better choice. Specifically,
> >
> > * **Reason for Normal Distribution:** Our initial choice of the Normal distribution was driven by **practicality and robustness**. We employ it not to perfectly fit the data density, but to derive a stable, efficient metric for "deviation" (z-score). In dynamic optimization scenarios, the Gaussian assumption offers computational simplicity and stable parameter estimation ($\mu, \sigma$), which helps avoid gradient instability.
> >
> > * **New Experiment with Beta Distribution:** Following your advice, we implemented a variant, **FaLW-beta**, replacing the Gaussian model with a Beta distribution. We estimated parameters $\alpha_c, \beta_c$ via MLE on the validation set and adapted our dynamic weight formulation using the Beta CDF, $F(p_i) = \text{BetaCDF}(p_i | \alpha_c, \beta_c)$. We mapped the probability to a deviation score $s_i \in [-1, 1]$:
> > $$
> > w_i = 1+\text{sign}(s_i)\cdot(|s_i|)^{1/\mathcal{B}_i}, \quad \text{where } s_i = 2\cdot (F(p_i)-0.5)
> > $$
> >
> > * **Results & Analysis:** We compared the two variants on CIFAR-100 and TinyImageNet (see table below). While FaLW-beta yields reasonable performance, it consistently underperforms FaLW-normal (higher Avg. Gap). Empirically, we observed that the Beta distribution's flexibility leads to **high sensitivity** and **erratic weight fluctuations** during unlearning, likely due to estimation errors in $\alpha, \beta$. In contrast, the Normal distribution acts as a robust proxy, providing smoother gradients and greater stability "out of the box." Thus, we retain the Normal distribution for its robustness but will include these Beta comparisons in the Appendix.
> >
> >     | Dataset (Setting) | Method | FA | RA  | TA  | MIA  | Avg. Gap $\downarrow$ |
> >     | :--- | :--- | :--- | :--- | :--- | :--- | :--- |
> >     | **CIFAR-100** | Retrain | 62.09 | 99.95 | 62.10 | 37.91 | 0.00 |
> >     | (ResNet-18, $\gamma=1/4$) | **FaLW-normal** | 61.74 | 99.98 | 62.69 | 38.26 | **0.33** |
> >     | (20% Forget) | FaLW-beta | 59.96 | 99.95 | 63.06 | 39.04 | 1.06 |
> >     | | | | | | | |
> >     | **TinyImageNet** | Retrain | 25.63 | 99.98 | 48.83 | 74.36 | 0.00 |
> >     | (VGG-16, $\gamma=1$) | **FaLW-normal** | 25.70 | 99.98 | 49.11 | 73.30 | **0.35** |
> >     | (10% Forget) | FaLW-beta | 26.91 | 99.97 | 48.77 | 72.09 | 0.91 |
> >
> >
> >
> > > **[Q1] How does this approach scale to a setup where we want to unlearn a particular class rather than unlearning a particular point from a given class.**
> >
> > We thank the reviewer for this insightful question, which touches upon the fundamental design premise and scope of our method.
> >
> > While FaLW is technically compatible with any gradient-based method using a forget set $\mathcal{D}_f$, the objectives of **Instance Unlearning** (our focus) and **Class Unlearning** are distinct:
> >
> > * **Random Instance Unlearning (Our Scope):** Here, the goal is **"Faithful-forgetting."** Even after forgetting specific samples in $\mathcal{D}_f$, the model *must* retain the generalized knowledge of that class learned from the retain set $\mathcal{D}_r$. In this context, **Over-forgetting** is a critical failure mode, as it implies the model has damaged its generalization capabilities for that class. FaLW is specifically engineered to prevent this by suppressing the unlearning weight ($w_i \to 0$) when a sample's prediction drops too low.
> >
> > * **Class Unlearning (Your Question):** In this scenario, the objective is the total erasure of a concept. Consequently, aggressively driving down the probability of that class (which we define as "Over-forgetting" in our instance-wise setting) is actually acceptable to some extent.
> >
> > Therefore, the gain from applying FaLW directly to the Class Unlearning task is not expected to exceed its gain from the Random Instance Unlearning task, as its core mechanism—designed to preserve class generalization—would likely counteract the goal of total class erasure. However, we agree that the "long-tailed" issue likely exists in class unlearning as well (e.g., forgetting rare classes vs. common classes). Adapting our deviation-aware framework to this new objective is a valuable direction for future research.

---

> > > ### Author Response · Authors · 2025-11-19
> > > **Part 3/3**
> > >
> > > > **[Q2] The paper describes FaLW as plug-and-play, but seems to have demonsrated it using only one specific unlearning approach. how can FaLW can be used in practice an any generic unlearning approach?**
> > >
> > > We thank the reviewer for this crucial question. **We explicitly designed FaLW as a generic, module-agnostic framework.**
> > >
> > > **Mechanism:** Theoretically, FaLW introduces a dynamic, instance-aware weight $w_i$ for each sample $(x_i, y_i)$ in the forget set. This weight is applied as a multiplicative factor to the base unlearning loss function $\mathcal{L}$:
> > > $$
> > > \mathcal{L} = w_i \cdot \mathcal{L_{base}}((x_i, y_i); \theta)
> > > $$
> > > Here, $\mathcal{L}_{base}$ can be the objective function of **any** gradient-based method (e.g., the negative log-likelihood in Gradient Ascent, or the cross-entropy with incorrect labels in Random Labeling). **FaLW acts as a "modulator"**, adjusting the intensity of these gradients based on the unlearning deviation, regardless of how the gradients are generated.
> > >
> > > **New Experiments:** While our initial submission followed the experimental setup of SalUn, we have conducted additional experiments to demonstrate FaLW's universality. We integrated FaLW with three other baselines: **Gradient Ascent (GA)**, **Random Labeling (RL)**, and **SFRon**. The results on CIFAR-100 (ResNet-18, $\gamma=1/4$, 20% forget ratio) are summarized below. As shown, introducing FaLW consistently reduces the **Avg. Gap** across all baselines.
> > >
> > > | Method | FA | RA | TA | MIA | Avg. Gap $\downarrow$ |
> > > | :--- | :--- | :--- | :--- | :--- | :--- |
> > > | Retrain | 62.09 | 99.95 | 62.10 | 37.91 | 0.00 |
> > > | GA | 95.22 | 96.66 | 68.83 | 4.78 | 19.07 |
> > > | **GA + FaLW** | 89.68 | 90.63 | 63.48 | 10.32 | **16.47** |
> > > | RL | 47.49 | 96.99 | 61.46 | 42.51 | 5.70 |
> > > | **RL + FaLW** | 62.24 | 99.97 | 66.46 | 37.75 | **1.17** |
> > > | SFRon | 65.30 | 99.98 | 66.42 | 33.69 | 2.95 |
> > > | **SFRon + FaLW** | 62.86 | 99.65 | 62.77 | 37.14 | **0.62** |
> > > | SalUn | 59.71 | 96.92 | 60.11 | 40.29 | 2.44 |
> > > | **SalUn + FaLW** | 61.74 | 99.98 | 62.69 | 38.26 | **0.33** |
> > >
> > > ---
> > >
> > > We hope that our response and the revised manuscript have satisfactorily addressed your concerns. We are happy to engage in further discussion if there are any remaining questions.
> > >
> > > Sincerely,
> > >
> > > The Authors of Submission 6959

---

> > > > ### Author Response · Authors · 2025-11-26
> > > >
> > > > Dear Reviewer V5ik:
> > > >
> > > > We sincerely appreciate the constructive feedback you provided, which has been crucial in improving our manuscript. We hope our response has satisfactorily addressed your concerns. We would be very grateful to hear your thoughts on our rebuttal and remain fully available for any further discussion.
> > > >
> > > > Best regards,
> > > >
> > > > The Authors of Submission 6959

---

### Author Response · Authors · 2025-12-01
**Summary of Reviewer Suggestion and Rebuttal Status**

Dear Area Chair,

First and foremost, we would like to express our sincere gratitude for your time and effort in handling our submission. We understand that taking over the assignment at this stage requires significant energy, and we deeply appreciate your dedication to the review process.
To facilitate your efficient evaluation of our work, we provide a summary of the current reviewer status and the major improvements made during the rebuttal.

### Current Reviewer Status
We have actively engaged with all reviewers to address their concerns. The current status is as follows:
* **Reviewer V5ik (Initial Score: 4):** Following our response and new experiments, this reviewer has replied stating that they have **no further questions** and are **willing to update their rating**.
* **Reviewers zD1B, N7tQ, PHd4 (Initial Score: 6):** We have provided point-by-point responses and extensive additional experiments to address their constructive comments. As they have not posted further inquiries, we trust that our revisions have effectively addressed their questions and reinforced their positive assessment.

### Key Revisions and Technical Highlights
To address the reviewers' comments comprehensively, we categorized the concerns into three main themes. We have significantly strengthened the paper in these areas:

**1. Proven "Plug-and-Play" Generalizability (Addressing V5ik, zD1B, N7tQ)**
* **Critique:** Reviewers requested evidence that FaLW works as a "Plug-and-Play" method.
* **Resolution:** We integrated FaLW with three additional distinct unlearning frameworks: **Gradient Ascent (GA)**, **Random Labeling (RL)**, and **SFRon**.
* **Outcome:** FaLW consistently reduced the Average Gap across **all** baselines, empirically proving its universality as a generic enhancement module.

**2. Justification of Modeling Assumptions: Normal vs. Beta (Addressing V5ik, PHd4)**
* **Critique:** Reviewers suggested that the assumption of normal distribution is too strong and does not satisfy the range of probabilities [0,1].
* **Resolution:** We implemented and compared a **Beta-distribution variant (FaLW-beta)** against our proposed **FaLW-normal**.
* **Outcome:** Experiments showed that **FaLW-beta** works well but **FaLW-normal outperforms FaLW-beta**. We found the Beta distribution to be highly sensitive to estimation errors in dynamic optimization, whereas the Normal distribution provides the necessary robustness and stability. This validates our design choice.

**3. Robustness and Stability Analysis (Addressing zD1B, N7tQ, PHd4)**
* **Critique:** Reviewers requested analysis on standard deviation, hyperparameter sensitivity, and extreme failure cases.
* **Resolution:**
    * **Standard Deviation:** Re-run experiments with 3 seeds confirm low variance (std $\approx 0.20$), proving stability.
    * **Sensitivity:** We added Appendix G analyzing hyperparameter $\tau$, identifying a stable region $[0.1, 0.2]$.
    * **Failure Mode:** We simulated extreme data scarcity (3 samples/class) to rigorously define the method's boundaries.

### Conclusion

We have successfully addressed the critical concerns from Reviewer V5ik, who is now willing to raise the rating. Additionally, we have strengthened the manuscript with extensive new experiments to satisfy the requests of the other positive reviewers. We are confident that our detailed responses and comprehensive additional experiments have satisfactorily resolved all concerns raised by the reviewers. We deeply appreciate your commitment to evaluating our work within this tight schedule and thank you for your valuable contribution to the ICLR community.

Sincerely,

The Authors of Submission 6959

---

### Meta-Review · Area_Chair_PpqT · 2026-01-07

**Summary:**

1. The paper's claim that FaLW is a plug-and-play, but only demonstrated on one unlearning method.
2.  The practicality is limited by a reliance on hard-to-obtain auxiliary data.
3. Lack of a formal certification guarantee.
4. he theoretical justification is weakened by an inappropriate choice of a Normal distribution to model probabilities, which is not naturally bounded between 0 and 1.

**Reviewer Concerns:**

Points 1 and 4 were addressed, and points 2 and 3 were partially addressed.

**Reviewer Scores:**

V5ik may raise the rating, and others remain the same.

---

### Decision · Program_Chairs · 2026-01-26

Accept (Poster)